# Mice with renal-specific alterations of stem cell-associated signaling develop symptoms of chronic kidney disease but surprisingly no tumors

**Adam Myszczyszyn**[1]*, **Oliver Popp**[2], **Severine Kunz**[3], **Anje Sporbert**[4], **Simone Jung**[1], **Louis C. Penning**[5], **Annika Fendler**[1], **Philipp Mertins**[2], **Walter Birchmeier**[1]

1 Cancer Research Program, Max Delbrück Center for Molecular Medicine in the Helmholtz Association (MDC), Berlin, Germany, 2 Proteomics, Max Delbrück Center for Molecular Medicine in the Helmholtz Association (MDC), Berlin, Germany, 3 Electron Microscopy, Max Delbrück Center for Molecular Medicine in the Helmholtz Association (MDC), Berlin, Germany, 4 Advanced Light Microscopy, Max Delbrück Center for Molecular Medicine in the Helmholtz Association (MDC), Berlin, Germany, 5 Faculty of Veterinary Medicine, Utrecht University, Utrecht, The Netherlands

* adam.myszczyszyn@gmail.com

**Data Availability Statement:** The mass spectrometry proteomics and phosphoproteomics data were deposited to the ProteomeXchange

## Abstract

Previously, we found that Wnt and Notch signaling govern stem cells of clear cell kidney cancer (ccRCC) in patients. To mimic stem cell responses in the normal kidney *in vitro* in a marker-unbiased fashion, we have established tubular organoids (tubuloids) from total single adult mouse kidney epithelial cells in Matrigel and serum-free conditions. Deep proteomic and phosphoproteomic analyses revealed that tubuloids resembled renewal of adult kidney tubular epithelia, since tubuloid cells displayed activity of Wnt and Notch signaling, long-term proliferation and expression of markers of proximal and distal nephron lineages. In our wish to model stem cell-derived human ccRCC, we have generated two types of genetic double kidney mutants in mice: Wnt-β-catenin-GOF together with Notch-GOF and Wnt-β-catenin-GOF together with a most common alteration in ccRCC, Vhl-LOF. An inducible Pax8-rtTA-LC1-Cre was used to drive recombination specifically in adult kidney epithelial cells. We confirmed mutagenesis of β-catenin, Notch and Vhl alleles on DNA, protein and mRNA target gene levels. Surprisingly, we observed symptoms of chronic kidney disease (CKD) in mutant mice, but no increased proliferation and tumorigenesis. Thus, the responses of kidney stem cells in the tubuloid and genetic systems produced different phenotypes, i.e. enhanced renewal versus CKD.

## Introduction

Over the past few decades, Wnt-β-catenin and Notch have emerged as fundamental signal transduction systems, which regulate maintenance and differentiation of stem cells in homeostasis and regeneration of adult organs [1, 2]. In the mouse kidney, Wnt and Notch signaling are upregulated in Sox9-positive stem/progenitor cells that contribute to regeneration of

Consortium via the PRIDE partner repository
(https://www.ebi.ac.uk/pride/) with the dataset
identifier PXD023491.

**Funding:** The study received financial and material
support from the Max Delbrück Center for
Molecular Medicine in the Helmholtz Association
(MDC), Berlin, Germany, and financial support
from the Urological Research Foundation (Stiftung
Urologische Forschung), Berlin, Germany. The
funders had no role in study design, data collection
and analysis, decision to publish, or preparation of
the manuscript. The authors received a salary from
the MDC, except for A.M. (doctoral position) and A.
F. (postdoctoral position) who received a salary/
scholarship from the Urological Research
Foundation, and except for L.C.P. who received a
salary from the Utrecht University, Utrecht, The
Netherlands. In addition, A.M. received a 3-month
salary (extension) from the MDC.

**Competing interests:** The authors have declared
that no competing interests exist.

proximal tubule, Loop of Henle and distal tubule [3]. Functionally, enhanced Notch accelerates Sox9-mediated repair. In addition, multipotent PROM1 (CD133)-positive stem/progenitor cells were isolated from both the tubular [4] and glomerular [5] segment of the human kidney, and were found to upregulate some components of Wnt and Notch [6]. PROM1 seems to play a role in activating Wnt in these cells [7]. Provocatively, a lineage tracing of Prom1-positive cells in the mouse kidney questioned their stem/progenitor cell functionality, as these cells displayed limited generative capacity in the postnatal kidney and were quiescent in the adult kidney [8]. In another study, unipotent progenitors specific for proximal, distal and collecting duct segments were reported in *Rainbow* mice, which drive homeostatic maintenance and regeneration of the kidney and are Wnt-responsive [9]. Moreover, deletion of the Wnt mediator β-catenin in kidney tubular cells and blocking Notch with a γ-secretase inhibitor delayed kidney recovery upon injury in mice [10, 11]. Together, activity of Wnt and Notch signaling seems to be a common feature of different stem/progenitor cell populations in the adult kidney.

Organoids are emerging as powerful tools to study properties of stem/progenitor cells in adult epithelial tissues. Organoids are 3D structures grown from stem/progenitor cells in culture with Matrigel and well-defined growth factors. During organoid formation, cells self-renew, differentiate and self-organize by cell sorting and spatially restricted lineage commitment in a manner similar to natural organs. Thus, organoids recapitulate stem/progenitor cell behaviors during homeostatic maintenance and regeneration of adult epithelia [12, 13]. Adult kidney organoids were created from normal human and mouse kidneys [14, 15]. However, characterization of stemness in these organoids has not been pushed forward to a satisfying level.

Clear cell renal cell carcinoma (ccRCC) is the most frequent and aggressive human kidney cancer [16]. Inactivation of the Von Hippel-Lindau (*VHL*) tumor suppressor gene followed by a HIF-1α- and HIF-2α-mediated hypoxic state occurs in the majority of ccRCC cases [17]. However, deletion of Vhl in mouse kidneys is insufficient for tumorigenesis [18]. Our laboratory identified CXCR4-, MET- and CD44-enriched cancer stem cells (CSCs) in human ccRCCs [19]. The CSCs displayed upregulation of Wnt and Notch signaling and quantitatively correlated with advanced stages, aggressiveness and metastasis of tumors. Treatment with small-molecule inhibitors of Wnt and Notch blocked self-renewal of the CSCs in patient-derived models, i.e. in subcutaneous and orthotopic xenografts, and in organoids and non-adherent spheres in cultures. This points to a crucial role of Wnt- and Notch-high stem/progenitor cells as actors in kidney tumorigenesis.

Upregulation of Wnt signaling in combination with genetic interferences at p53 or p21 in mouse kidneys using embryonic Cres was sufficient to overcome Wnt-induced p53-p21-mediated senescence and resulted in development of malignant lesions that displayed characteristics of non-ccRCCs [20, 21]. Concomitant upregulation of Wnt and K-ras signaling in embryonic mouse kidneys also drove formation of malignant lesions that resembled non-ccRCCs, including pediatric tumors [22, 23]. Hyperactivation of Notch signaling in combination with Vhl loss in adult mouse kidneys contributed to development of premalignant lesions with some hallmarks of ccRCC [24, 25]. However, upregulation of either Wnt or Notch in adult Cd133-positive kidney cells in mice failed to produce tumors [8]. Answer to a fundamental question remains elusive on whether the functional interplay of Wnt and Notch drives tumorigenesis in the kidney, also in the context of Vhl inactivation.

To reveal the importance of Wnt and Notch signaling for adult renal cells with stem/progenitor-like functionality *in vitro*, without enriching populations positive for a specific marker, we have established tubular organoids (tubuloids) from whole mouse kidneys in Matrigel in serum-free conditions. Then, to mimic development of human ccRCC from Wnt- and Notch-dependent stem cells, we have generated genetic mutants in mice, i.e. Wnt-β-catenin-GOF

together with Notch-GOF or with Vhl-LOF, using an adult kidney tubular epithelium-specific Cre to circumvent embryonic or postnatal kidney defects and non-renal confounding influences.

## Materials and methods

### Mice

The Landesamt für Gesundheit und Soziales (LaGeSo) in Berlin approved the mouse study (G0342/13). Animal experiments were conducted in accordance with European, national, federal state and institutional regulations. Mice were housed and bred in the IVC cages type II at 21˚C ambient temperature, at 50–60% humidity, in a 12-h dark/light cycle and in pathogen-free conditions. LC1-Cre mice [26] were a kind gift from Klaus Rajewsky (MDC). β-catenin-loxP [27], Notch-loxP [28] and LacZ-loxP [29] mice came from in-house colonies. Pax8-rtTA B6.Cg-Tg(Pax8-rtTA2S*M2)1Koes/J [30] and Vhl-loxP B6.129S4(C)-Vhltm1Jae/J [31] mice were purchased from the Jackson Laboratory. The studies in tubuloids and genetic mouse models were performed using both male and female mice of a mixed C57BL/6J and FVB/NJ background, unless otherwise stated. To establish the genetic mouse models, adult one-month-old Pax8-rtTA-LC1-Cre-loxP mice and controls carrying loxP alleles, but no Pax8-rtTA and/or LC1-Cre (both males and females) were exposed for 5 days to doxycycline (Sigma-Aldrich) at the concentration of 0.2 mg/ml in drinking water supplemented with 5% sucrose. To alleviate suffering, mice were checked at least twice per week according to a score sheet until illness symptoms were observed. Sick mice were immediately anesthesized with iso-flurane and sacrificed by cervical dislocation, then biological material was collected.

### Preparation of kidney cell suspensions

Kidneys were harvested and transferred to MEM with Earle's Salts (with 2.2 g/l NaHCO3, without L-Glutamine) medium (Biochrom) supplemented with 10% FCS (Life Technologies), 1x Non-Essential Amino Acids (NEAA, Thermo Fisher Scientific), 2 mM L-Glutamine (Bio-chrom), 1x Penicillin-Streptomycin (Life Technologies) and 125 μg/ml Amphotericin B (Bio-mol). Whole kidneys were minced into pieces smaller than 1 mm3, digested in 2 mg/ml Collagenase P (Roche) for 45 min and filtered through sieves with 70 and 40 μm pore size. Erythrocytes were lysed in RBC lysis buffer and lymphocytes were depleted by MACS using CD45 microbeads (Miltenyi Biotec) according to the manufacturer's instructions.

### Kidney tubuloid cultures

Freshly isolated cell suspensions were seeded at $10^5$ cells per well in 25 μl Matrigel (75%, Corning) lenses in 48-well plates. Matrigel drops were overlayed with 250 μl tubuloid medium (Table 1). Medium was changed every other day. Tubuloids were passaged for the first time after 14 days and subsequently once per week. Matrigel lenses were mechanically broken and tubuloids were collected and dissociated to small cell clusters using TrypLE Select (Life Technologies) for 10 min. In the first two passages, collected tubuloids were additionally filtered through a sieve with 40 μm pore size. Cultures were passaged in 1:3 ratios. If single cell suspensions were needed, tubuloids were dissociated in TrypLE for 1 hr. Cell clusters from tubuloids were cryopreserved at an early passage in 1 ml Recovery Cell Culture Freezing Medium (Thermo Fisher Scientific) according to the manufacturer's instructions. BrdU (Thermo Fisher Scientific) treatment (at 10 μM) was performed for 4 hrs prior to tubuloid collection for staining. For functional assays, 100 ng/ml human Wnt3a (R&D Systems) was added. For the examination of structural changes of tubuloids upon R-spondin-1 withdrawal or ICG-001 treatment,

**Table 1. Medium components for tubuloid culture.**

| Component | Function | Concentration | Supplier |
|---|---|---|---|
| Advanced DMEM/F12 with GlutaMax | Basal medium | – | Life Technologies |
| Penicillin-Streptomycin | Antibiotics | 1x | Life Technologies |
| Amphotericin B | Antifungal | 125 μg/ml | Biomol |
| HEPES | Buffer | 10 mM | Life Technologies |
| N-acetylcysteine | Antioxidant | 1.25 mM | Sigma-Aldrich |
| B27 | Serum-free growth supplement | 1x | Life Technologies |
| N2 | Serum-free growth supplement | 1x | Life Technologies |
| Nicotinamide | Increasing organoid formation and lifespan via inhibiting sirtuin activity involved in apoptosis and differentiation [32, 33] | 10 mM | Sigma-Aldrich |
| Human EGF | Stimulating proliferation, inhibiting apoptosis [32, 33] | 50 ng/ml | Peprotech |
| A83-01 | Blocking differentiation and promoting long-term culture of stem cells via inhibiting TGFβ signaling [32, 33] | 500 nM | Sigma-Aldrich |
| Human R-spondin-1 | Maintaining stem cells via agonizing Wnt signaling [32, 33] | 500 ng/ml | Peprotech |
| Hydrocortisone (HC) | Acting anti-inflammatory via repressing activity of NF-κB and AP-1 [34], inducing Yap signaling [35] | 0.5 μg/ml | Sigma-Aldrich |
| Prostaglandin E2 (PGE2) | Preventing anoikis and promoting stem cell survival via enhancing Wnt signaling [32] | 1 μM | Sigma-Aldrich |

single cells were seeded at 45,000 cells in 48-well plates and cultured 7 days in tubuloid medium without R-spondin-1 or cultured 24h in normal medium and 6 days in medium containing ICG-001 (Biochempartner) at IC50. Brightfield and fluorescence images were acquired with a Leica DMI 6000 B microscope using the Leica LAS X software.

## Proteomics and phosphoproteomics

Three independent replicates of early (OEP) and long-term (OLP) passage tubuloids were released from Matrigel by mechanical disruption, collected in a conical tube and incubated with the non-enzymatic Cell Recovery Solution (Corning) according to the manufacturer's instructions to remove Matrigel rests. Three independent replicates of Epcam-positive kidney epithelial cells (control kidney, CK) were MAC-sorted from freshly isolated single cell suspensions from whole mouse kidneys using microbeads (Miltenyi Biotec) according to the manufacturer's instructions. Global proteomic and phosphoproteomic characterization of the samples was achieved by the tandem mass tag (TMT)-based quantitation using the TMT 10-plex reagents (Thermo Fisher Scientific), as previously described [36]. Briefly, samples were lysed in urea lysis buffer containing protease and phosphatase inhibitors, subjected to tryptic in-solution digest and labeled with the TMT 10-plex reagents. After pooling, samples were separated into 24 fractions for the proteomic and 12 fractions for the phosphoproteomic analysis using basic reversed-phase HPLC. Phosphopeptides were enriched from each fraction using robot-assisted iron-based IMAC on an AssayMap Bravo system (Agilent). Proteomic and phosphoproteomic samples were measured on a Q Exactive HF-X orbitrap mass spectrometer (Thermo Fisher Scientific) connected to an EASY-nLC system (Thermo Fisher Scientific) applying a 110-min online HPLC gradient. MS acquisition was performed at a resolution of 60,000 in a scan range from 350 to 1500 Th. Data-dependent MS2 scans were carried out at a resolution of 45,000 with an isolation window of 0.7 Th and a maximum injection time of 86 ms using the top 20 peaks from each precursor scan for HCD fragmentation. For the data

analysis, the MaxQuant software package version 1.6.10.43 [37] was used. A FDR cutoff of 0.01 was applied for peptides and proteins and database search was performed using the mouse Uniprot database (July 2018), including isoforms. Variable modifications included phosphorylation on serine, threonine and tyrosine, methionine-oxidation, acetylated N-termini and deamidation of asparagine and glutamine, while the TMT10-plex reporter ion quantitation was turned on using a PIF setting of 0.5. Log2-transformed and median-MAD centered corrected reporter ion intensities were used for quantitation. Protein groups were filtered for proteins, which were identified by at least one unique peptide and at least two peptides in total and had valid values across all samples. Phosphorylation site tables were filtered for valid values across all samples. For significance calling, a two-sample moderated t-testing (limma R package) [38] was applied. A multiple comparison correction was done using the Benjamini-Hochberg method (adjusted P-values). The KEGG PATHWAY (mmu04310) and a previously published list [39] served as a combined reference protein list for the Wnt signaling heatmap. The KEGG PATHWAY (mmu04330) served as a reference protein list for the Notch signaling heatmap. A previously published list [40] served as a reference protein list for the Yap signaling heatmap.

## Section staining

Tubuloids were released from Matrigel by mechanical disruption, collected in a conical tube and fixed in 10% (v/v) Neutral-buffered Formalin overnight. Tubuloids were transferred to a 2-ml tube and overlayed with 1.5% agarose. The resulting agarose beads were dehydrated and embedded in paraffin. All stainings were performed on 5-μm sections. For haematoxylin-eosin (H&E) staining, sections were rehydrated and stained with haematoxylin (Fluka) for 4 min and with eosin (Merck) for 2 min. For immunofluorescence, sections were rehydrated, and antigen retrieval in TRIS-EDTA and blocking with 10% donkey serum (Bio-Rad) were performed. Sections were incubated overnight with rat anti-BrdU (1:100, Abcam, ab6326), mouse anti-E-cad (1:300, BD Biosciences, 610181), rabbit anti-Laminin (1:500, Abcam, ab11575), rat anti-Zo-1 (1:200, Santa Cruz, sc-33725), mouse anti-MDR1 (1:50, Santa Cruz, sc-55510) and mouse anti-AQP3 (1:50, Santa Cruz, sc-518001) primary antibodies or with Biotinylated LTL (1:300, Vector Laboratories, B-1325) and Rhodamine-conjugated PNA (1:300, Vector Laboratories, RL-1072) lectins, followed by incubation with donkey fluorescent dye-conjugated Alexa488 anti-mouse, Alexa488 anti-rabbit, Alexa488 anti-rat and Cy3 anti-rat (1:250, Jackson) secondary antibodies or with Alexa647-conjugated Streptavidin (1:200, Thermo Fisher Scientific) and by counterstaining with DAPI (0.2 μg/ml, Thermo Fisher Scientific) for 1 hr. For immunohistochemistry, sections were rehydrated, antigen retrieval in TRIS-EDTA was performed, endogenous peroxidase activity was blocked in 3.5% (v/v) hydrogen peroxide and blocking with 10% donkey serum (Bio-Rad) was performed. Sections were incubated overnight with rabbit anti-Pax8 primary antibody (1:2000, Proteintech, 10336-1-AP), followed by incubation with peroxidase-conjugated donkey anti-rabbit secondary antibody from a kit (Dako) for 1 hr. Sections were developed using the substrate-chromogen system (Dako) and counterstained with haematoxylin (Fluka) for 2 min. Kidneys and spleens were fixed in 10% (v/v) Neutral-buffered Formalin overnight, dehydrated and embedded in paraffin. All stainings were performed on 5 μm sections. For immunohistochemistry, sections were rehydrated, antigen retrieval in TRIS-EDTA was performed, endogenous peroxidase activity was blocked in 3.5% (v/v) hydrogen peroxide and blocking with 10% donkey serum (Bio-Rad) was performed. Sections were incubated overnight with rabbit anti-Sox9 (1:1000, Abcam, ab185966), rabbit anti-N1icd (1:100, Cell Signaling, 4147S), rabbit anti-Ki67 (1:200, Thermo Fisher Scientific, RM-9106-S), mouse anti-phospho-γH2AX (1:500, Merck Millipore, 05–636), mouse anti-p53 (1:100, Santa Cruz, sc-126) and mouse anti-p21 (1:100, Santa Cruz, sc-6246) primary

antibodies, followed by incubation with peroxidase-conjugated donkey anti-rabbit and anti-mouse secondary antibodies from a kit (Dako) for 1 hr. Sections were developed using the substrate-chromogen system (Dako) and counterstained with haematoxylin (Fluka) for 2 min. For PAS staining, sections were rehydrated, stained using the Periodic Acid-Schiff (PAS) Kit (Sigma-Aldrich) according to the manufacturer's instructions and counterstained with haematoxylin (Fluka) for 2 min. For LacZ staining, cryosections were prepared. Kidneys were placed in 30% sucrose in PBS overnight and embedded in OCT compound (Sakura Finetek). 10-μm sections were fixed in 0.2% (v/v) glutaraldehyde for 10 min, washed three times in washing solution (PBS, 2 mM MgCl2, 0.01% sodium deoxycholate and 0.02% NP-40), incubated overnight in staining solution (PBS, 2 mM MgCl2, 0.01% sodium deoxycholate, 0.02% NP-40, 5 mM K3Fe(CN)6, 5 mM K4Fe(CN)6 and 1 mg/ml X-gal (Roth)) and counterstained with nuclear fast red (Merck) for 5 min. Images were taken with an Axio Imager.Z1 microscope (for immunofluorescence) or an Axio Scope.A1 microscope (for H&E staining, PAS staining, LacZ staining and immunohistochemistry) using the Zen software (all Zeiss). Selected immunofluorescence images were processed with Fiji ImageJ.

## Whole-mount staining

Tubuloids were released from Matrigel by mechanical disruption, collected, transferred to a 2-ml tube, fixed in 10% (v/v) Neutral-buffered Formalin for 1.5 hrs and blocked in 10% donkey serum (Bio-Rad) for 2 hrs. Tubuloids were incubated overnight with mouse anti-E-cad primary antibody (1:300, BD Biosciences, 610181), Biotinylated LTL lectin (1:300, Vector Laboratories, B-1325) and Rhodamine-conjugated PNA lectin (1:300, Vector Laboratories, RL-1072), and then overnight with donkey Alexa488 anti-mouse secondary antibody (1:250, Jackson), Alexa647-conjugated Streptavidin (1:200, Thermo Fisher Scientific) and DAPI (0.2 μg/ml, Thermo Fisher Scientific). Tubuloids were taken up in 1.5% low-melting agarose and transferred to chamber slides. Confocal z-stacks were acquired with a LSM700 or LSM710 microscope (Zeiss) using a long working distance C-Achroplan 32x/0.85 water immersion objective and Immersol W (Zeiss). DAPI, Alexa488, Cy3 and Alexa647 fluorophores were excited with a 405 nm, 488 nm, 561 nm and 633 nm laser, respectively, and detected by sequential scanning using BP filters 410–460 nm, 500–540 nm, 565–620 nm and 640–720 nm, respectively. Images were acquired with a pixel size of 0.116 or 0.142 μm (lateral) and 1.11 or 1.16 μm (axial) with 12-bit and line average 2. If necessary, detector gain and laser power were slightly adjusted with z-depth to keep a constant signal. Restoration of confocal z-stacks was performed with the Huygens Deconvolution software (SVI) running on a dedicated data processing workstation (Acquifer) using a CMLE algorithm with 60 iterations and a SNR of 12. After deconvolution, confocal z-stacks were 3D reconstructed using the Imaris 9.1 software (Bitplane/Oxford Instruments) and either the volume reconstruction and clipping planes or orthogonal sections were used to visualize and explore the 3D structure of tubuloids.

## Transmission electron microscopy

Tubuloids were released from Matrigel by mechanical disruption, collected in a conical tube and fixed in 2% (w/v) formaldehyde and 2.5% (v/v) glutaraldehyde in 0.1 M phosphate buffer for 2 hrs. After embedding in 10% agarose in a 2-ml tube, samples were post-fixed with 1% (v/v) osmium tetroxide (Sigma-Aldrich), dehydrated in a graded series of EtOH and embedded in PolyBed® 812 resin (Polysciences). Ultrathin sections (60–80 nm) were stained with uranyl acetate (Polysciences) and lead citrate (Sigma-Aldrich) and examined at 80 kV with an EM 910 electron microscope (Zeiss). Acquisition was performed with a Quemesa CCD camera and the iTEM software (Emsis).

## TUNEL assay

Kidneys were fixed in 10% (v/v) Neutral-buffered Formalin overnight, dehydrated and embedded in paraffin. 5-μm sections were rehydrated, stained using the ApopTag Plus Peroxidase In Situ Apoptosis Detection Kit (Merck Millipore) according to the manufacturer's instructions and counterstained with haematoxylin (Fluka) for 2 min. Images were taken with an Axio Scope.A1 microscope using the Zen software (Zeiss).

## Immunoblotting

Whole kidneys were homogenized and protein was extracted from 20 mg of tissue in 1 ml RIPA buffer supplemented with protease inhibitors (Roche) for 2 hrs. 10 μg of protein was separated by SDS-PAGE using a 10% separating and a 4.5% stacking gel and transferred to a PVDF membrane with 0.2 μm pore size (Roth) via a semidry transfer. The membrane was blocked with 5% skim milk in TBS-T, cut in half and probed with the mouse anti-β-catenin (1:1000, BD Transduction Laboratories, 610153) or mouse anti-α-Tubulin (1:10000, Sigma-Aldrich, T9026) antibody overnight and with the peroxidase-conjugated donkey anti-mouse antibody (1:5000, Jackson) for 1 hr. The antibodies were diluted in 5% BSA in TBS-T. Bands were visualized using the Western Lightning Plus-ECL substrate (PerkinElmer) for 5 min and a Fusion SL imaging system (Vilber).

## Real-time qPCR

Total RNA from tubuloids and whole mouse kidneys was isolated using TRIzol (Invitrogen), and 1 μg/20 μl of RNA was reversely transcribed to cDNA using MMLV reverse transcriptase (Promega) according to the manufacturer's instructions. Real-time qPCR reactions were performed in a CFX96-C1000 thermal cycler (Bio-Rad) with the PowerUp SYBR Green Master Mix (Applied Biosystems) and 10 μM exon-exon junction-spanning primers, using a standard protocol with 44 cycles according to the manufacturer's instructions. Primer sequences are listed in the Table 2. Primer specificity was tested by melting curve analyses and running reactions with a negative control (without cDNA). Relative mRNA expression values were normalized to the endogenous control Gapdh using the 2-ΔΔCt method.

## Genomic DNA PCR

Genomic DNA was isolated from whole kidneys using the GeneJET Genomic DNA Purification Kit (Thermo Fisher Scientific) according to the manufacturer's instructions. β-catenin and Vhl alleles were amplified using 100 ng of DNA, 10 μM primers and a standard protocol with 50 and 40 cycles and annealing temperature of 65˚C and 56˚C, respectively. A forward primer 5'-3': GGTACCTGAAGCTCAGCG−CACAGCTG and a reverse primer 5'-3': ACGTGTGGC AAGTTCCGCGTCATCC were used to amplify the β-catenin allele [27]. A forward primer 1 5'-3': CTGGTACCCAC−GAAACTGTC, a forward primer 2 5'-3': CTAGGCACCGAGCTTAGAGG TTTGCG and a reverse primer 5'-3': CTGACTTCCACTGATGCTTGTCACAG were used to amplify the Vhl allele [41].

## Drug response assay

Single cells were seeded at 15,000 cells per well in 9 μl Matrigel (75%, Corning) lenses in 96-well plates and overlayed with 100 μl tubuloid medium. After 24 hrs, medium was replaced with fresh medium containing ICG-001 (Biochempartner) or LF3 (Selleckchem). Medium was changed every other day. CellTiterGlo assay (Promega) was performed after 6 days according to the manufacturer's instructions.

**Table 2. Primer sequences for real-time qPCR.**

| Gene | Forward primer 5'-3' | Reverse primer 5'-3' |
| --- | --- | --- |
| Krt8 | CTCAAAGGCCAGAGGGCATC | TTAATGGCCATCTCCCCACG |
| Krt18 | ACTGGTCTCAGCAGATTGAGG | CCGAGGCTGTTCTCCAAGTT |
| Cldn4 | CTTCATCGGCAGCAACATCG | GATGACCATAAGGGCTCGGG |
| Abcb1b | AGTGGCTCTTGAAGCCGTAA | AAACTCCATCACCACCTCACG |
| Slc3a1 | AAAATGCCTTGACTGGTGGCA | CCTCAACAGCGTATCTGAAGTCT |
| Slc40a1 | GAGCCAGTGTCCCCAACTAC | CTTGCAGCAACTGTGTCACC |
| Aqp3 | ATCGTTGTGGGGAGATGCTT | ACCAAGATGCCAAGGGTGAC |
| Atp6ap2 | GGCAAAACAAGAGAACACCCA | CCAAGGCCAAGCCGATCATA |
| Wnk1 | CCTCAAGTATGGCACAGGGG | GCTGTATTCCCTGCTGCTGA |
| Dkk1 | CGGGGGATGGATATCCCAGA | ACGGAGCCTTCTTGTCCTTTG |
| Axin2 | GCGCTTTGATAAGGTCCTGG | TCATGTGAGCCTCCTCTCTTTT |
| Cyclin D1 | CTGGATGCTGGAGGTCTGTGA | AGGGGGTCCTTGTTTAGCCAG |
| Myc | TTGGAAACCCCGCAGACAG | GCTGTACGGAGTCGTAGTCG |
| Hey1 | GAGCGTGAGTGGGATCAGTG | GCTTAGCAGATCCCTGCTTCT |
| Hes1 | CTGGTGCTGATAACAGCGGA | GGAATGCCGGGAGCTATCTT |
| Hey3 (Heyl) | GTCTTGCAGATGACCGTGGA | CGGGCATCAAAGAACCCTGT |
| Ca9 | GTCATTGGAGCTATGGAGG | CTCATAACCCAGAAGTTCCAG |
| Hk2 | GTGACAGACAATGGTCTCCAGAG | GCCAGGCATTCGGCAATG |
| Pdk1 | GCAGTTCCTGGACTTCG | CAATCTAACAGGCAACTCTTG |
| Ldha | GATGGATCTCCAGCATGGCAG | GTGATAATGACCAGCTTGGAGTTCG |
| Glut1 | GCTATAACACTGGTGTCATCAACG | CGTGGTGAGTGTGGTGGATG |
| Vegfa | CAGGCTGCTGTAACGATGAA | TTTCTTGCGCTTTCGTTTTT |
| Ngal | ACAACCAGTTCGCCATGGTA | AAGCGGGTGAAACGTTCCTT |
| Kim-1 | CAGGGTCTCCTTCACAGCAG | CCACCACCCCCTTTACTTCC |
| Gsta1 | AGCCCGTGCTTCACTACTTC | CAATCTCCACCATGGGCACT |
| Il1b | TTCAGGCAGGCAGTATCA | CCAGCAGGTTATCATCATCA |
| Cd11b | CCACACTAGCATCAAGGGCA | GCTTCACACTGCCACCGT |
| Ly6c | GCAGTGCTACGAGTGCTATGG | ACTGACGGGTCTTTAGTTTCCTT |
| Icam1 (Cd54) | GTCCGCTGTGCTTTGAGAAC | GAGGTCCTTGCCTACTTGCT |
| Cd68 | ACTTCGGGCCATGTTTCTCT | GCTGGTAGGTTGATTGTCGT |
| Ccr2 (Cd192) | GCCATCATAAAGGAGCCATACC | ATGCCGTGGATGAACTGAGG |
| Vim | TGGATCAGCTCACCAACGAC | AAGGTCAAGACGTGCCAGAG |
| Fn1 | ATGAGAAGCCTGGATCCCCT | GGAAGGGTAACCAGTTGGGG |
| a-sma | ACATCAAGGAGAAGCTGTGCT | TTTCGTGGATGCCCGCTG |
| Col1a1 | CATGAGCCGAAGCTAACCCC | GGGTTTCCACGTCTCACCAT |
| Col3a1 | AGTGGGCATCCAGGTCCTAT | GGGTGAAAAGCCACCAGACT |

## Blood assays

Blood was collected from hearts of freshly sacrificed mice to tubes coated with 0.5 M EDTA pH 8 (Invitrogen) to prevent clotting. BUN concentrations were measured using the i-STAT 1 System with the CHEM8+ cartridges for patient testing (Abbott) according to the manufacturer's instructions. For plasma collection, cells and platelets were pelleted from the blood by centrifugation for 15 min at 2000 g. EPO concentrations in supernatants (plasma) were measured using the Quantikine Mouse Erythropoietin Immunoassay (R&D Systems) according to the manufacturer's instructions.

### Descriptive statistics and significance testing

The Pearson correlation matrix (coefficient) was calculated by the cor function of the base R package between all samples of mouse kidney epithelia (control kidney, CK), early passage tubuloids (OEP) and long-term passage tubuloids (OLP) based on normalized log2 intensity values for 9,000 proteins and 16,000 phosphorylation sites detected in the mass spectrometry analysis. The matrix was plotted using the pheatmap R package. For the functional annotation clustering of the most upregulated proteins using DAVID bioinformatic tool (DAVID Bioinformatics Resources 6.8, NIAID/NIH), 9,000 proteins detected in the proteomic analysis were subjected to both the 0.1% FDR cutoff (adjusted P-value < 0.001) and log2 fold change cutoff of > 0.5 for both OEP over CK and OLP over CK. 722 proteins were selected. Using the entire mouse (Mus musculus) proteome as a background for the UNIPROT_ACCESSION identifier, 671 proteins (DAVID IDs) were identified and analyzed. Defined DAVID defaults were used and the Enrichment Thresholds (EASE Scores, modified Fisher-exact P-values) and adjusted P-values (Benjamini-Hochberg correction) for all terms in each cluster were subjected to the cutoff of < 0.001. 10 clusters were determined. The overall enrichment score for each cluster was calculated based on the Enrichment Thresholds for all terms. The most enriched term was selected to represent each cluster based on the lowest adjusted P-value. All other statistical analyses were performed in GraphPad Prism (GraphPad). All data are presented as mean ± SD (error bars), unless otherwise stated. To assess the normal distribution of the data, the Shapiro-Wilk test ($\alpha = 0.05$) was performed. The unpaired, two-tailed Student's t-test and the ordinary one-way ANOVA followed by the Dunnett's multiple comparison were used to analyze data, which passed the normality test. To compare groups, which did not pass the normality test, the alternative non-parametric Kruskal-Wallis test followed by the Dunn's multiple comparison were performed. The ordinary one-way ANOVA fol-lowed by the Dunnett's multiple comparison were used to analyze the real-time qPCR data on tubuloids. A P-value < 0.05 was considered statistically significant. IC50 values were calculated by a non-linear regression analysis of the response and log10 of the inhibitor concentration fitting a curve with a variable slope (four parameters). Statistical details of the experiments can be found in the figure legends.

## Results

### Long-term tubuloid cultures from adult mouse kidneys

We isolated single epithelial cells from whole adult mouse kidneys and established 3D tubuloid cultures with high efficiency in Matrigel and serum-free conditions (n = 20). The protocol was adapted to specific needs of kidney cells with stem/progenitor-like functionality (Table 1). Within 14 days of culture, single cells grew continuously and formed tubuloids with sizes up to 1.5 mm in diameter (Fig 1A). By H&E staining, we classified three tubuloid types: cystic tubuloids with one or more cell layers and a single lumen (with few tubuloids containing one or two additional smaller lumina), solid filled tubuloids, and alveolar tubuloids with multiple lumina, accounting for 65%, 25% and 10% (Fig 1B).

Tubuloid formation from single cells was five times higher in the first passage in comparison to freshly seeded cultures (Fig 1C and 1D), indicating self-renewal capacity of cells. For maintenance, tubuloids were serially passaged starting from small cell clusters (Fig 1E), which allowed to culture them for at least 3.5 months (passage 12, S1A Fig). Tubuloid cultures were established from mice with a mixed C57BL/6J and FVB/NJ background, but they could also be generated from C57BL/6J mice with similar efficiency (S1B Fig). Hence, we created long-term tubuloid cultures from single adult mouse kidney epithelial cells, which predominantly

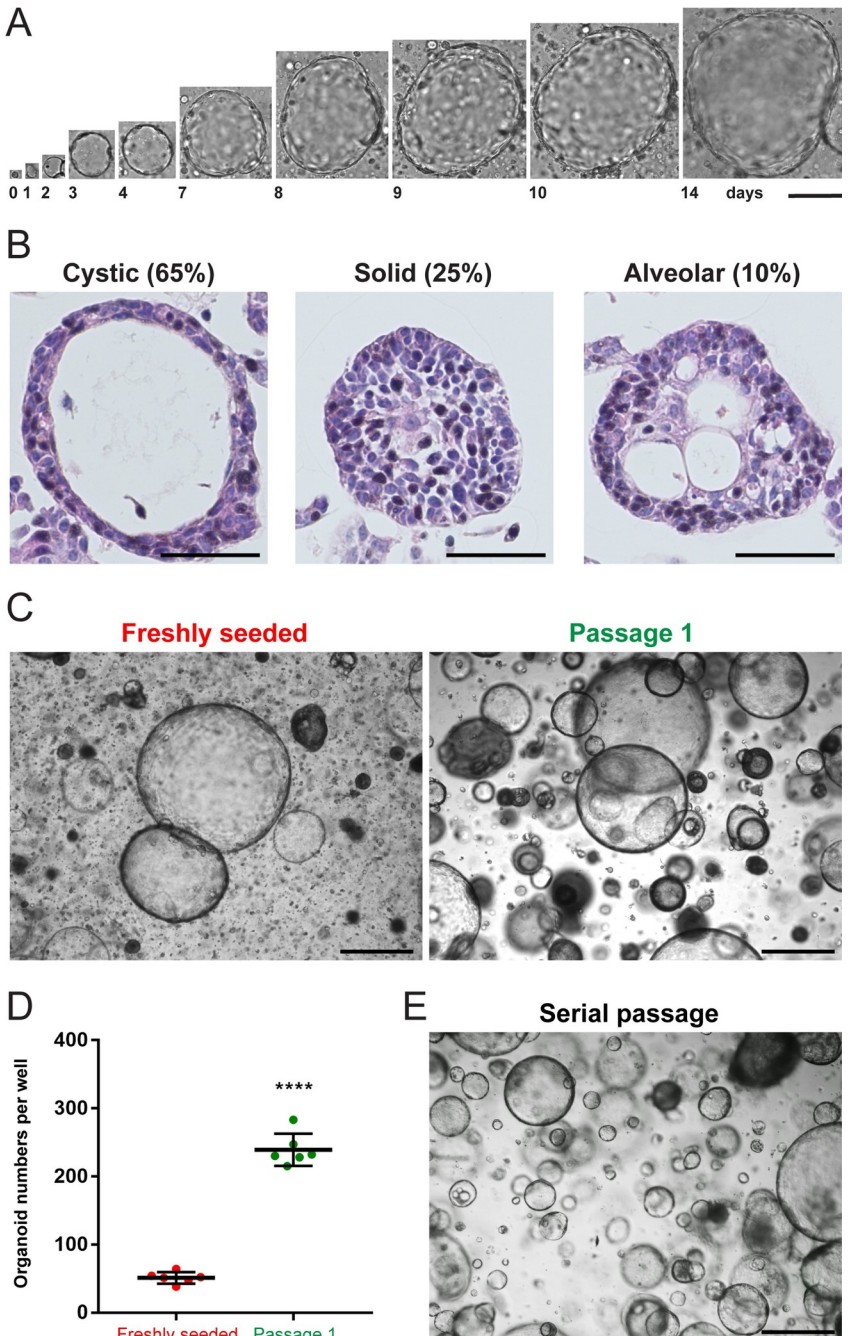

**Fig 1. Long-term tubuloid cultures from adult mouse kidneys.** (A) Brightfield images acquired on consecutive days of an tubuloid formed from a single freshly seeded cell. (B) H&E staining showing freshly seeded tubuloids with cystic (left), solid (middle) and alveolar (right) morphology. (C) Brightfield images of a freshly seeded (left) and a once passaged (right) tubuloid culture grown from $10^5$ single cells. (D) Quantification of tubuloid numbers in a freshly seeded and passage 1 culture. (E) Brightfield image of a serially passaged tubuloid culture after 5 passages. Data information: scale bars, 100 μm in A, 50 μm in B, 500 μm in C and E. 20 freshly seeded tubuloid cultures were established in total. In A, one independent replicate was examined. For quantification in B, 180 tubuloids were counted in total. One independent replicate was examined. For quantification in C and D, $10^5$ cells were seeded and tubuloids with diameters $\geq$ 100 μm were counted after 14 (freshly seeded cells) or 7 (passaged cells) days of culture (6 wells, technical replicates). One independent replicate was examined. In D, the graph shows mean ± SD (error bars). Data passed the Shapiro-Wilk normality test (α = 0.05). The unpaired, two-tailed Student's t-test was performed; P-value, < 0.0001, ****p < 0.0001. In E, three independent replicates were examined.

exhibited cystic morphologies, consistent with epithelial organoid cultures derived from other adult organs [13].

## Examination of kidney tubuloids by deep proteomic and phosphoproteomic analyses revealed that the cultures closely resembled adult mouse kidney epithelia and were phenotypically stable over time

We performed mass spectrometry-based deep bulk proteomics and phosphoproteomics. Three independent replicates of adult kidney epithelial cells (control kidney), early passage tubuloids (passage 2, 1-month culture) and long-term passage tubuloids (passage 12, 3.5-month culture) were compared. To prepare control samples, Epcam-positive kidney epithelial cells were MAC-sorted from freshly isolated single cell suspensions from whole mouse kidneys, similar to a protocol for preparing adult human kidney tubuloids [15]. We could detect 9,000 proteins and 16,000 phosphorylation sites of these proteins, which is the maximum coverage currently possible [36]. We calculated Pearson correlations between the samples and found high correlations, at least 0.8, between adult kidney epithelial cells (control kidney, CK) and both early passage tubuloids (OEP) and long-term passage tubuloids (OLP) for both proteome (Fig 2A) and phosphoproteome (Fig 2B). This indicated that tubuloids closely resembled kidney epithelia. Moreover, OLP recapitulated OEP for both proteome (Fig 2A) and phosphoproteome (Fig 2B), as shown by correlations of around 1, confirming that tubuloid cultures were phenotypically stable over time.

## Kidney tubuloids displayed enhanced proliferation, and upregulation of stem cell-associated Wnt, Notch and Yap signaling

We also recorded the importance of many upregulated proteins in tubuloids (both OEP and OLP) in comparison to kidney epithelial cells. Using a bioinformatic tool for functional annotation clustering, DAVID, we discovered 10 biological processes in tubuloids, which were governed by upregulated proteins (Fig 2C). The top three enriched processes were cell cycle, ribosomal activity and DNA replication, which indicated the presence of proliferating cells. The proteins involved included the typical proliferation markers Ki67, Cyclin D1 (Ccnd1) and Pcna (Fig 2D). In line with the increased tubuloid-forming capacity, enhanced proliferation suggested enrichment in cells with stem/progenitor-like functionality in tubuloids and thus the recapitulation of the potential of the adult kidney for renewal. DAVID analysis also revealed enrichment in epithelial cell polarity-related proteins of adherens junctions and basement membranes (Fig 2C).

By manual curation, proteomic analysis in both OEP and OLP revealed upregulation of components of stem cell-associated Wnt (Fig 2E) and Notch (Fig 2F) signaling. Upregulated Wnt components included ligands Wnt4, Wnt7b and Wnt10a, co-receptors Lrp5 and Lrp6, the destruction complex disassembly-mediating protein Dvl2 and targets Axin2, Birc5, Cd44 and Cyclin D1 (Ccnd1). Upregulated Notch components included the ligand Jag2, receptors Notch1, Notch2 and Notch3, the S2 cleavage-mediating metalloproteinase Adam17, S3 cleavage γ-secretase complex-related Aph1a and Ncstn as well as co-activators Rbpj, Maml1, Maml2 and Kat2b. Proteomic analysis also revealed in both OEP and OLP upregulation of another stem cell-associated signaling system, Yap (S2 Fig). Upregulated Yap components were direct Yap targets. Thus, tubuloids and adult mouse kidney epithelia largely shared proteome and phosphoproteome patterns, which remained unchanged over passages. Moreover, strong proliferation of tubuloids and upregulation of the Wnt, Notch and Yap signaling systems were the signs of the enrichment in cells with stem/progenitor-like functionality.

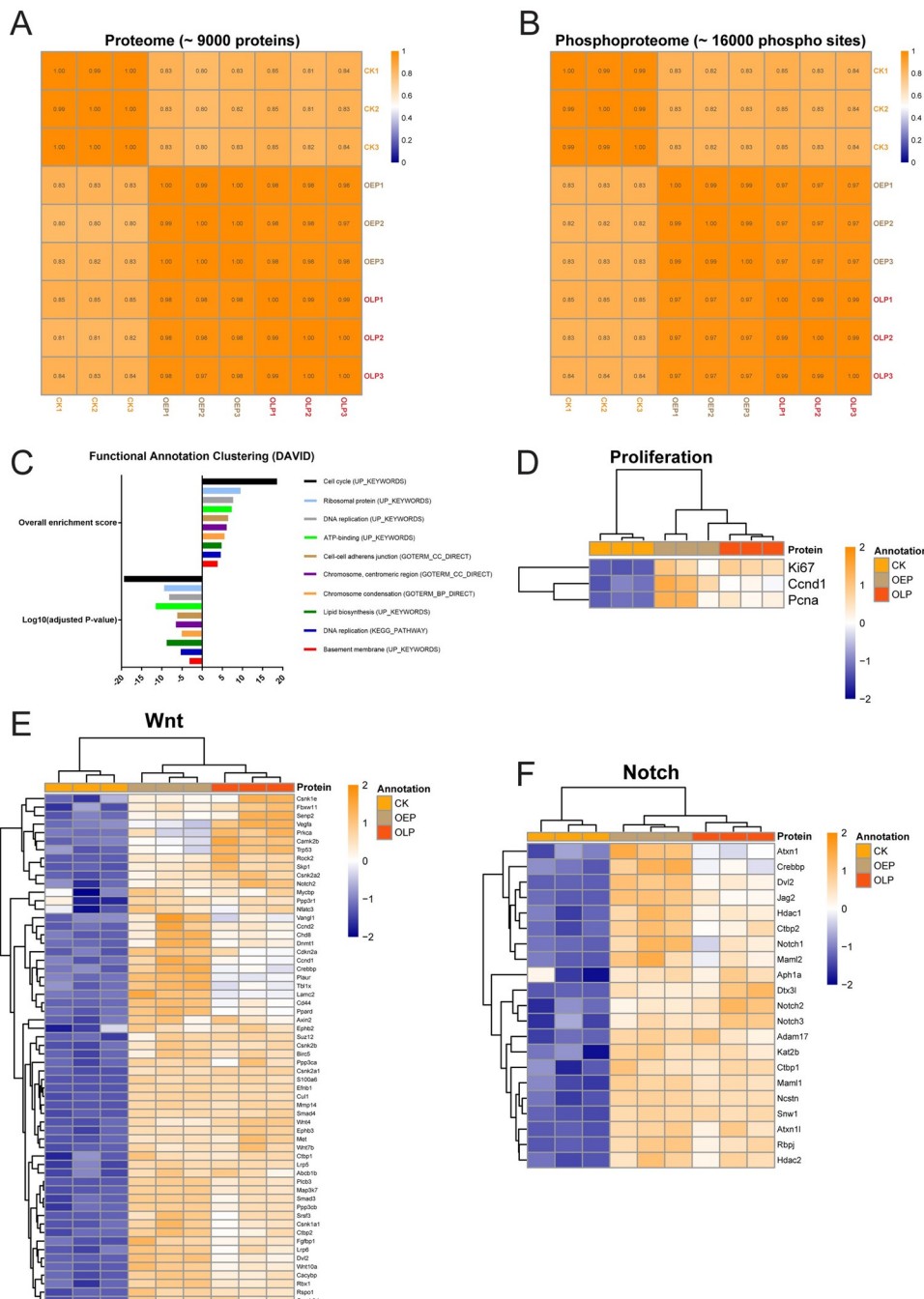

**Fig 2. Kidney tubuloids closely resemble adult mouse kidney epithelia, are phenotypically stable over time, and display enhanced proliferation and stem cell-associated Wnt and Notch signaling.** (A) Heatmap for the Pearson correlation matrix (coefficient) of the expression levels of 9,000 proteins between all samples of mouse kidney epithelia (control kidney, CK), early passage tubuloids (OEP) and long-term passage tubuloids (OLP). The experimental columns were correlated against each other. (B) Heatmap for the Pearson correlation matrix (coefficient) of the phosphorylation levels of 16,000 corresponding phosphorylation sites between all samples of mouse kidney epithelia (control kidney, CK), early passage tubuloids (OEP) and long-term passage tubuloids (OLP). The experimental columns were correlated against each other. (C) Functional annotation clustering of the most upregulated proteins in tubuloids using DAVID bioinformatic tool. Shown are 10 most enriched clusters, based on the overall enrichment scores, with their representative terms (biological processes), based on the adjusted P-values. (D) Proteomic heatmap for the typical proliferation markers Ki67, Cyclin D1 (Ccnd1) and Pcna in both early (OEP) and long-term (OLP) passage tubuloids in comparison to mouse kidney epithelia (control kidney, CK). (E) Proteomic heatmap for components of Wnt signaling in both early (OEP) and long-term (OLP) passage tubuloids in comparison to mouse

kidney epithelia (control kidney, CK). (F) Proteomic heatmap for components of Notch signaling in both early (OEP) and long-term (OLP) passage tubuloids in comparison to mouse kidney epithelia (control kidney, CK). Data information: in D-F, the heatmaps show normalized log2 intensity values for three independent replicates of CK, OEP and OLP. A 5% FDR (adjusted P-value < 0.05) cutoff and a log2 fold change cutoff of > 0 were applied for both OEP over CK and OLP over CK. The values were scaled (z-score by row) with breaks from $\leq$ -2 to $\geq$ 2.

Further, we examined protein expression in OEP and OLP of three markers associated with human and mouse resident renal stem/progenitor cells, i.e. Prom1 [3–5], Sox9 [3, 42] and Pax2 [4]. However, Prom1 was downregulated in OEP and OLP, as compared to CK (S3A Fig). We detected Sox9 neither in OEP and OLP nor in CK due to either a technical issue related to coverage or a very low expression. Alternatively, we did not find the presence of Sox9 in OEP and OLP using immunohistochemistry (S3B Fig). Moreover, a slight downregulation of Pax2 in OEP and OLP was found, as compared to CK (S3A Fig). Surprisingly, proteomic analysis revealed in tubuloids (mainly in OEP) upregulation of Six2, a marker for an embryonic population of nephron progenitor cells [43] (S3A Fig). This might suggest that tubuloid-forming cells are a result of a de-differentiation process, but further studies to confirm a functional role of Six2-positive cells in tubuloids are necessary. We also detected in OEP and OLP increased levels of Met and Cd44, which are the markers of cancer stem cells that we identified in human clear cell kidney tumors [19]. Both are also Wnt targets (Fig 2E).

## Kidney tubuloids contained differentiated and polarized tubular epithelial cells

Ubiquitous staining of Pax8 in nuclei (marked in brown) confirmed that tubuloid cells were of kidney origin (Fig 3A). Tubuloids contained differentiated kidney epithelial cells, as indicated by the presence of adherens junctions between lateral membranes of neighboring cells, which were positive for E-cadherin (E-cad, marked in orange, Fig 3B), and by upregulation of Keratin 8 (Krt8), Claudin 4 (Cldn4) and *Keratin 18 (Krt18)* (Fig 3C). In line with DAVID analysis, cells displayed pronounced epithelial polarity, as shown by the location of Laminin in the basal laminae (marked in green) and of Zo-1-positive tight junctions at the apical sides (marked in red) (Fig 3D).

Cells were positive either for Lotus Tetragonolobus Lectin (LTL, marked in red), a marker of proximal tubular epithelial cells, or Peanut Agglutinin (PNA, marked in green), a marker of epithelial cells of distal nephrons, i.e. distal tubules and collecting ducts, and both markers were detected within the same tubuloids (Fig 3E). Proteomic analysis revealed enrichment in 48 membrane transporters in both OEP and OLP (Fig 3F), including known transporters of proximal tubular cells such as amino acid transporters Slc1a4 and Slc1a5, the glucose transporter Slc2a1, the myo-inositol transporter Slc5a11, ion transporters Slc12a6 and Slc26a2, zinc transporters Slc30a6 and Slc39a10 and efflux transporters Abcb1b, Abcc3 and Abcc4. In addition, known transporters of distal nephrons were upregulated, including Slc6a9, a glycine transporter of distal and connecting tubules, and ion transporters of collecting ducts, Slc4a1ap, Slc4a7, Atp6ap2 and Trpv4. Proton transporters Atp6v0a1 and Atp6v0a2, which are present in both proximal tubules and collecting ducts, were also enriched. Proteomic analysis also revealed upregulation of Wnk1, a common cytoplasmic marker (a kinase) of distal and connecting tubules and of cortical collecting ducts, and Gata3, a nuclear marker (a transcription factor) of collecting ducts (Fig 3G). In LTL/PNA-double-positive tubuloids, we confirmed in a subset of LTL-positive cells expression of Abcb1b (marked in cyan), another marker of proximal tubular cells, which did not overlap with PNA-positive cells of distal nephrons (Fig 3H). In these tubuloids, we also detected in a subset of PNA-positive cells expression of Aquaporin

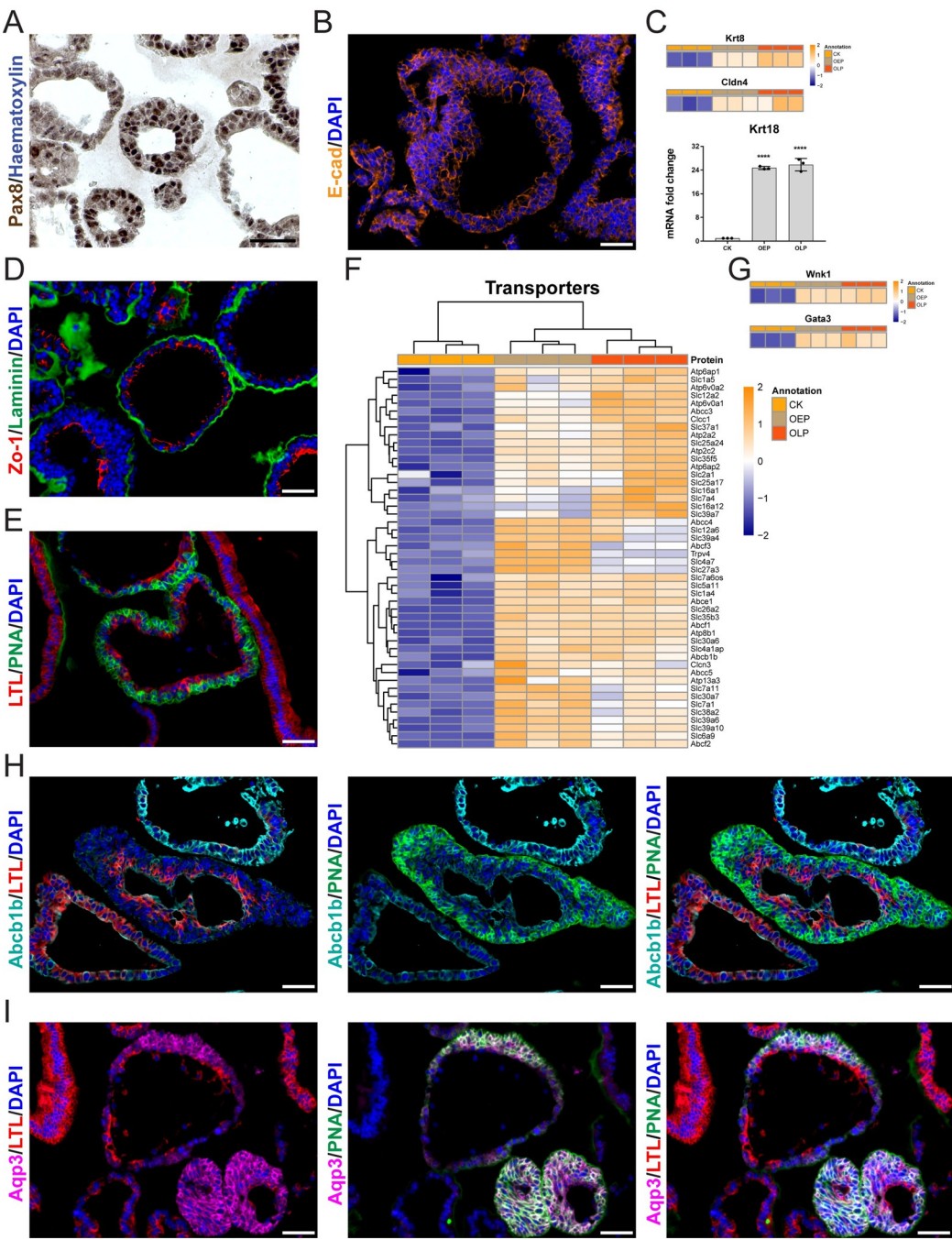

**Fig 3. Kidney tubuloids exhibit markers of differentiated and polarized tubular epithelial cells.** (A) Immunohistochemistry for Pax8 (brown) of an tubuloid culture. (B) 2D immunofluorescence for E-cad (orange) of an tubuloid culture. (C) Proteomic heatmaps for Krt8 and Cldn4 in both early (OEP) and long-term (OLP) passage tubuloids in comparison to mouse kidney epithelia (control kidney, CK; upper part), and real-time qPCR analysis of gene expression of Krt18 in both early (OEP) and long-term (OLP) passage tubuloids in comparison to whole mouse kidney cells (control kidney, CK; lower part). (D) 2D immunofluorescence for Laminin (green) and Zo-1 (red) of an tubuloid culture. (E) 2D immunofluorescence for LTL (red) and PNA (green) of an tubuloid culture. (F) Proteomic heatmap for 48 membrane transporters in both early (OEP) and long-term (OLP) passage tubuloids in comparison to mouse kidney epithelia (control kidney, CK). (G) Proteomic heatmaps for Wnk1 and Gata3 in both early (OEP) and long-term (OLP) passage tubuloids in comparison to mouse kidney epithelia (control kidney, CK). (H) 2D immunofluorescence for Abcb1b (cyan), LTL (red) and PNA (green) of an tubuloid culture. (I) 2D immunofluorescence for Aqp3 (magenta), LTL (red) and PNA (green) of an tubuloid culture. Data information: scale bars in A/B/D/E/H/I, 50 μm. In A, nuclei are counterstained with haematoxylin; in B/D/E/H/I, nuclei are counterstained with DAPI. In C, the graph shows mean ± SD (error bars). The

ordinary one-way ANOVA followed by the Dunnett's multiple comparison to CK were performed; both P-values, 0.0001, ****p < 0.0001. In C/F/G, the heatmaps show normalized log2 intensity values for three independent replicates of CK, OEP and OLP. A 5% FDR (adjusted P-value < 0.05) cutoff and a log2 fold change cutoff of > 0 were applied for both OEP over CK and OLP over CK. Values were scaled (z-score by row) with breaks from ≤ -2 to ≥ 2. In A/B/D/E/H/I, three independent replicates were examined. For the real-time qPCR analysis of Krt18 in C, three independent replicates in technical triplicates were examined.

3 (Aqp3, marked in magenta), a marker of principal cells of collecting ducts, which was mutually exclusive with LTL-positive cells (Fig 3I). We also observed mutually exclusive expression of LTL and PNA in solid and alveolar tubuloids (S4A Fig). Whole-mount confocal microscopy confirmed expression of LTL, PNA and E-cad in 3D-reconstructed cystic tubuloids (S4B and S4C Fig and S1 Video).

Transmission electron microscopy confirmed epithelial cell polarity and complexity of tubuloids. Cystic tubuloids developed microvilli (MV) at apical membranes towards the lumen (L), a typical feature of proximal tubular cells (S5A Fig). At the basal side, microvilli-free basal laminae (arrowhead) and filopodia (F) were seen (S5A Fig). Junctional complexes (JC) between neighboring cells, which are involved in barrier formation, were present at lateral membranes at the luminal side. JC included from the apical to basal side: tight junctions (zonula occludens, black asterisk), adherens junctions (belt desmosomes, zonula adhaerens, white asterisk), spot desmosomes (macula adhaerens, red asterisks) and closely aligned or nearly fused lateral membranes (white and black arrowheads, respectively) (S5B Fig). At the basal side, we observed spot desmosomes (red asterisk) and closely aligned or nearly fused lateral membranes (white and black arrowheads, respectively) (S5C Fig). Endocytic events, i.e. clathrin-coated pits with diameters of 100 nm, which are typical for proximal tubular cells [44], were observed at apical and lateral membranes (marked by arrowheads, S5D Fig). Together, apart from cells with stem/progenitor-like functionality, kidney tubuloids contained differentiated tubular epithelial cells with apical and basal polarity and complex intercellular junctions.

## Wnt signaling controlled formation, growth and differentiation of kidney tubuloids

We examined the dependence of tubuloids on Wnt signaling by removing or adding single Wnt activators and by blocking Wnt with small-molecule inhibitors. Withdrawal of the Wnt activator R-spondin-1 reduced tubuloid numbers (tubuloid-forming capacity) and sizes. Addition of the Wnt ligand Wnt3a did not improve tubuloid yield (Fig 4A and 4B), possibly due to autocrine supply of Wnt4, Wnt7b or Wnt10a. Treatments with ICG-001, an inhibitor that blocks β-catenin-Tcf-mediated transcription, and with LF3, another β-catenin-Tcf inhibitor developed in our laboratory [45] (Fig 4C), produced concentration-dependent inhibition of tubuloid growth (Fig 4D and 4F) and concentration-dependent decrease in cell viability (Fig 4E and 4G). By H&E staining, we observed upon R-spondin-1 removal a switch from predominant cystic to fully solid tubuloids, while treatment with ICG-001 at IC50 did not result in morphological changes (Fig 4H). Transition to solid morphologies induced by R-spondin-1 withdrawal prompted to study the importance of Wnt signaling for the control of tubuloid differentiation. By real-time qPCR, we examined in tubuloids upon R-spondin-1 removal or ICG-001 treatment expression of selected genes, which are induced in differentiated kidney tubular epithelial cells. We found increases in abundance of *Krt8*, *Krt18* and *Cldn4* (Fig 4I). We also observed upregulation of proximal tubule genes *Abcb1b*, *Slc3a1* and *Slc40a1* (Fig 4J) as well as of distal nephron genes *Aqp3*, *Atp6ap2* and *Wnk1* (Fig 4K). Immunofluorescence staining confirmed upregulation of Aqp3 in these tubuloids (Fig 4L). Altogether, mouse

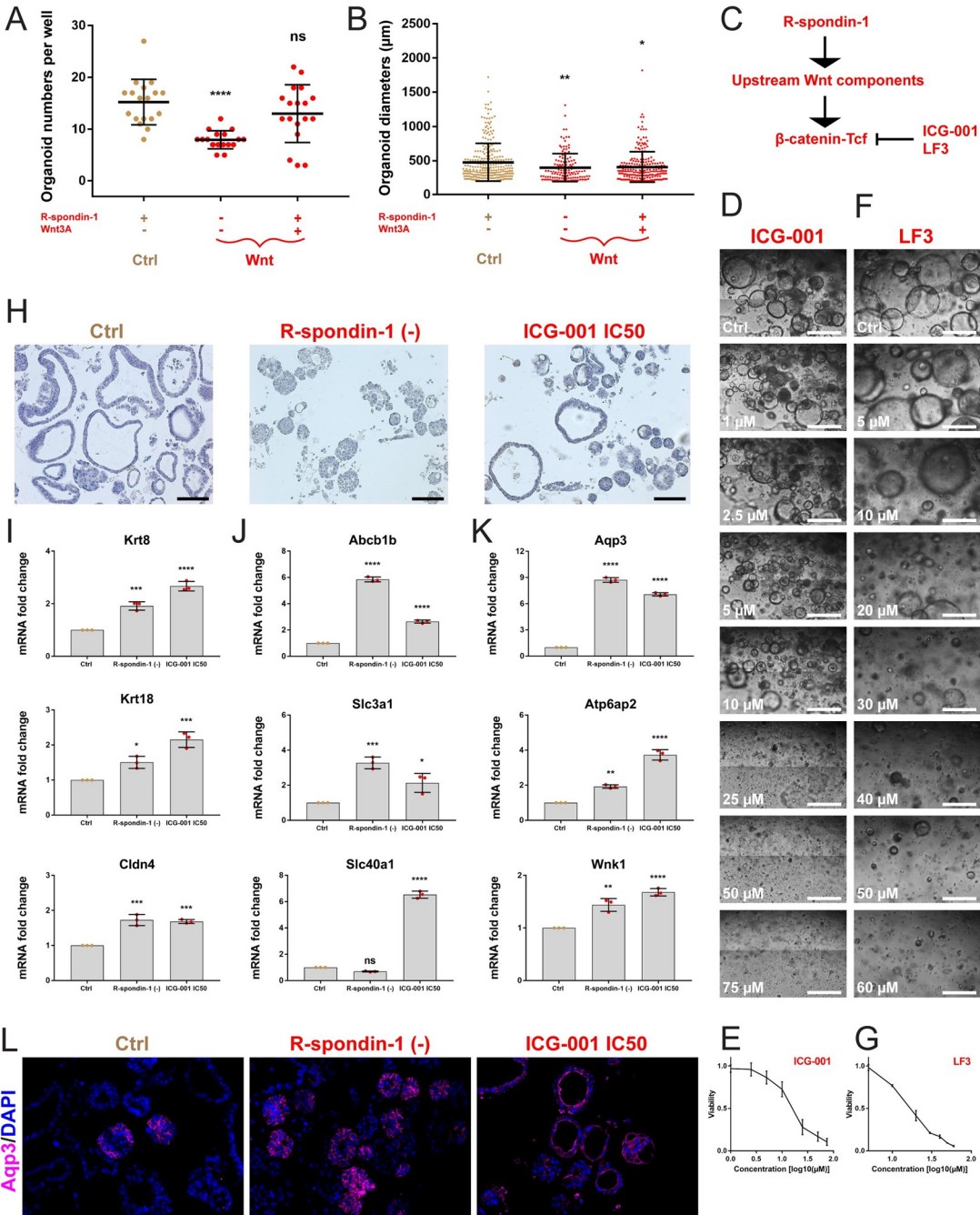

**Fig 4. Wnt signaling controls kidney tubuloid formation, growth and differentiation.** (A) Tubuloid numbers per well with indicated growth factor combinations (withdrawal or addition). (B) Diameters (μm) of tubuloids (single dots) from wells with indicated growth factor combinations (withdrawal or addition). (C) Scheme of Wnt signaling with a common target for the small-molecule inhibitors ICG-001 and LF3. (D and F) Brightfield images of tubuloid cultures at indicated ICG-001 (D) and LF3 (F) concentrations (μM) on day 6 of treatment. (E and G) Curves showing the dependence of tubuloid viability (fold change ATP luminescence) on log10 ICG-001 (E) and LF3 (G) concentration (μM) on day 6 of treatment. (H) H&E staining of tubuloid cultures upon R-spondin-1 removal or treatment with ICG-001 at IC50. (I) Real-time qPCR analysis of gene expression of differentiation markers Krt8, Krt18 and Cldn4 in tubuloids upon R-spondin-1 removal or treatment with ICG-001 at IC50. (J) Real-time qPCR analysis of gene expression of markers of proximal tubular cells, Abcb1b, Slc3a1 and Slc40a1, in tubuloids upon R-spondin-1 removal or treatment with ICG-001 at IC50. (K) Real-time qPCR analysis of gene expression of markers of distal nephrons, Aqp3, Atp6ap2 and Wnk1, in tubuloids upon R-spondin-1 removal or treatment with ICG-001 at IC50. (L) 2D immunofluorescence for Aqp3 (magenta) in tubuloids upon R-spondin-1 removal or treatment with ICG-001 at IC50. Data information: in A and B, culture medium containing DMEM/F12 with GlutaMax, HEPES, N-acetylcysteine, B27, N2,

nicotinamide, hydrocortisone (HC) and prostaglandin E2 (PGE2) is common to all the conditions. For quantification, $10^5$ cells were freshly seeded and tubuloids with diameters $\geq$ 200 μm were counted after 14 days of culture. The graphs show mean ± SD (error bars). In A, data passed the Shapiro-Wilk normality test (α = 0.05). The ordinary one-way ANOVA followed by the Dunnett's multiple comparison to complete tubuloid medium (Ctrl, first from the left) were performed; P-values from left to right, 0.0001, 0.2132; ****p < 0.0001, ns: non-significant. In B, data did not pass the Shapiro-Wilk normality test (α = 0.05). The alternative non-parametric Kruskal-Wallis test followed by the Dunn's multiple comparison to complete tubuloid medium (Ctrl, first from the left) were performed; P-values from left to right, 0.0090, 0.0129; **p < 0.01, *p < 0.05. In A and B, three independent replicates with 6 technical replicates (wells) were examined and collectively shown. Ctrl in D-G, tubuloids cultured in complete medium and treated with DMSO at re-spective concentrations. Scale bars in D and F, 500 μm. In E and G, the curves show mean ± SD (error bars). In E, ICG-001 produced the IC50 of 14.66 μM; in G, LF3 produced the IC50 of 15.72 μM. In D-G, three independent replicates in technical triplicates (wells) were examined. Ctrl in H-L, tubuloids cultured in complete medium and treated or not with DMSO at a respective concentration. Scale bars in H, 100 μm. Three independent replicates were examined. In I-K, the graphs show mean ± SD (error bars). The ordinary one-way ANOVA followed by the Dunnett's multiple comparison to Ctrl were performed; P-values from left to right in I; 0.0004, 0.0001 for Krt8; 0.0153, 0.0002 for Krt18; 0.0002, 0.0003 for Cldn4; in J: 0.0001, 0.0001 for Abcb1b; 0.0005, 0.0164 for Slc3a1; 0.0910, 0.0001 for Slc40a1; in K: 0.0001, 0.0001 for Aqp3; 0.0014, 0.0001 for Atp6ap2; 0.001, 0.0001 for Wnk1; ****p < 0.0001, ***p < 0.001, **p < 0.01, *p < 0.05, ns: non-significant. Three independent replicates in technical triplicates were examined. Scale bars in L, 50 μm. Three independent replicates were examined.

kidney tubuloid cultures required Wnt signaling to promote formation, growth and differenti-ation into proximal and distal nephron lineages.

## Generation of double kidney mutants in mice to model stem cell-driven development of human ccRCC

To investigate a relation between Wnt- and Notch-high cells with stem/progenitor-like func-tionality enriched in our mouse kidney tubuloids and CSCs in human ccRCC, we genetically modified Wnt and Notch signaling in mice. We generated two types of double mouse mutants by the loxP-mediated recombination of adult kidney epithelial cells: i) Wnt-β-catenin gain of function combined with Notch1 intracellular domain (N1icd) gain of function, hereafter called β-catenin-GOF; Notch-GOF, and ii) Von Hippel-Lindau loss of function combined with Wnt-β-catenin gain of function, hereafter called Vhl-LOF; β-catenin-GOF. For the details of the generation of the genetic system and mutants, see Fig 5A. Vhl-LOF; Notch-GOF mutant mice had already been established using different Cre systems [24, 25].

To examine cellular patterns of recombination, we established Pax8-rtTA(+); LC1-Cre(+); LacZ(+) reporter mice, which carried a loxP-flanked stop cassette upstream of the LacZ sequence encoding β-galactosidase. After doxycycline treatment of one-month-old mice, immunohistochemistry revealed Cre expressed in nuclei of kidney epithelial cells in both corti-cal and medullary parts, including proximal tubular cells (Fig 5B). The Cre pattern was con-firmed by cytoplasmic LacZ staining after treatment with X-gal. Glomeruli (marked by black lines) remained negative for Cre and LacZ. We did not observe Cre- and LacZ-positive cells in doxycycline-treated Pax8-rtTA(-); LC1-Cre(+); LacZ(+) control kidneys, indicating that the genetic system was not leaky. Then, we treated with doxycycline critical one-month-old Pax8-rtTA(+); LC1-Cre(+); β-catenin loxP/wt; Notch loxP/wt (β-catenin-GOF; Notch-GOF) mice and Pax8-rtTA(+); LC1-Cre(+); Vhl loxP/loxP; β-catenin loxP/wt (Vhl-LOF; β-catenin-GOF) mice (n = 20 per line). We also treated controls carrying β-catenin- and Notch- or Vhl- and β-catenin-loxP alleles, but no Pax8-rtTA or LC1-Cre (n = 20).

We examined in kidney mutants the recombination of β-catenin, Notch and Vhl alleles on genomic DNA, protein and mRNA target gene levels. We detected DNA bands of recombined (700 bp) and wild-type (900 bp) β-catenin alleles in both mutants, as compared to control kid-neys with only wt alleles (Fig 5C). In Vhl-LOF; β-catenin-GOF mutant, we detected DNA bands of recombined (260 bp) and non-recombined loxP (460 bp) Vhl alleles, as opposed to control kidneys with only loxP alleles (Fig 5D). Recombined β-catenin protein accumulated in

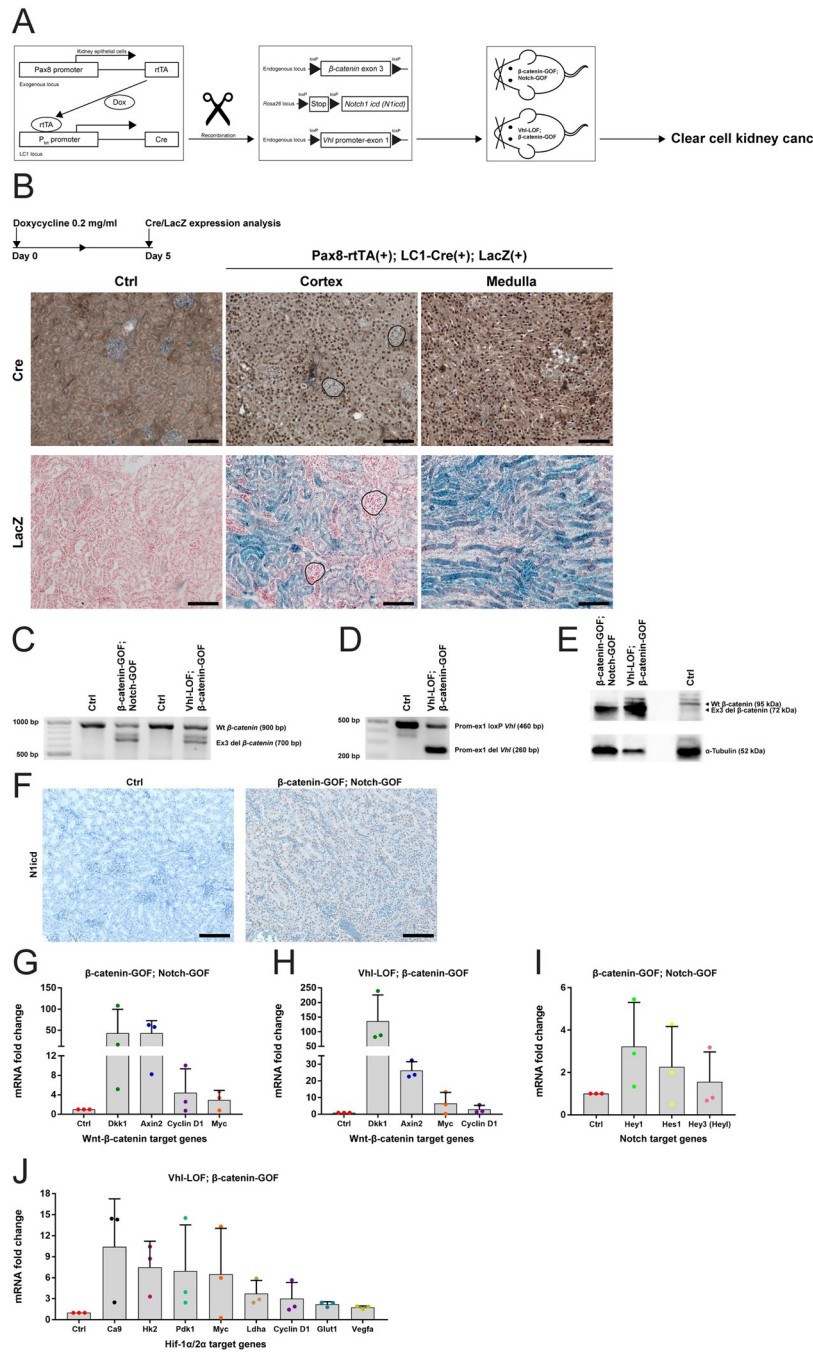

**Fig 5. Generation of double kidney mutants in mice.** (A) Scheme of the genetic system to generate mutant kidneys.
(B) Time course and pattern of recombination in kidneys of Pax8-rtTA(+); LC1-Cre(+); LacZ(+) reporter mice. (C)
PCR for DNA recombination of the loxP-flanked exon 3 sequence of β-catenin gene in β-catenin-GOF; Notch-GOF
and Vhl-LOF; β-catenin-GOF mutant kidneys (700 bp) versus controls (900 bp). (D) PCR for DNA recombination of
the loxP-flanked promoter-exon 1 sequence of Vhl gene in Vhl-LOF; β-catenin-GOF mutant kidneys (260 bp) versus
controls (460 bp). (E) Immunoblotting for recombination of β-catenin in β-catenin-GOF; Notch-GOF and Vhl-LOF;
β-catenin-GOF mutant kidneys (72 kDa) versus controls (95 kDa). (F) Immunohistochemistry for nuclear N1icd in β-
catenin-GOF; Notch-GOF mutant cortical kidneys versus controls. (G) Fold change of expression of selected β-catenin
target genes in β-catenin-GOF; Notch-GOF mutant kidneys versus controls. (H) Fold change of expression of selected
β-catenin target genes in Vhl-LOF; β-catenin-GOF mutant kidneys versus controls. (I) Fold change of expression of
selected Notch target genes in β-catenin-GOF; Notch-GOF mutant kidneys versus controls. (J) Fold change of
expression of selected Hif-1α/2α target genes in Vhl-LOF; β-catenin-GOF mutant kidneys versus controls. Data
information: abbreviation in A; dox, doxycycline. Scale bars in B, 100 μm. In the upper panel, nuclei are counterstained

with haematoxylin; in the lower panel, nuclei are counterstained with nuclear fast red. Glomeruli are marked by black lines. Two independent replicates of Pax8-rtTA(+); LC1-Cre(+); LacZ(+) mice and one of Pax8-rtTA(-); LC1-Cre(+); LacZ(+) controls were examined. Abbreviations in C-E; wt, wild-type; ex3 del, exon 3 deletion; prom-ex1 loxP/del, promoter-exon 1 loxP/deletion. Scale bars in F, 100 μm. Nuclei are counterstained with haematoxylin. In C-F, three independent replicates per line were examined. In G-J, graphs show mean + SD (error bars). Three independent replicates (mice, dots) in technical triplicates were analyzed. No statistical test was performed because of high inter-mouse variance. However, increase trends were observed.

both mutants (72 kDa band), as compared to controls with only wt protein (95 kDa band) (Fig 5E). We observed nuclear N1icd in β-catenin-GOF; Notch-GOF mutant, but not in control kidneys (Fig 5F). We found upregulation of classical target genes of β-catenin, i.e. *Dkk1*, *Axin2*, *Cyclin D1* and *Myc*, in both β-catenin-GOF; Notch-GOF (Fig 5G) and Vhl-LOF; β-catenin-GOF (Fig 5H) mutant kidneys, as opposed to controls. Expression of classical target genes of Notch, i.e. *Hey1*, *Hes1* and *Hey3*, in β-catenin-GOF; Notch-GOF mutant kidneys (Fig 5I), and expression of classical target genes of Hif-1α/2α (upregulated upon Vhl loss), i.e. *Ca9*, *Hk2*, *Pdk1*, *Myc*, *Ldha*, *Cyclin D1*, *Glut1* and *Vegfa*, in Vhl-LOF; β-catenin-GOF mutant kidneys (Fig 5J) also increased, as compared to controls. Thus, our data show that in mice targeted mutagenesis in adult kidney epithelial cells occurred and that mutant cells were not cleared over time.

## No tumorigenesis was observed in double mutant kidneys

Mutant mice were examined 1–8 months after the doxycycline pulse and exhibited severe illness symptoms. These included reduced body weight (Fig 6A) and enlarged spleens, called splenomegaly (Fig 6B and 6D). Mutant kidneys appeared to be of smaller sizes (Fig 6B), but their relative weight did not change (Fig 6C). Kidneys and spleens of mutant mice were pale, suggesting anemia (Fig 6B).

We performed Periodic acid-Schiff (PAS) staining of mutant kidneys, which detects tissues with high proportions of sugar macromolecules such as tubular and glomerular basement membranes and brush borders of proximal tubules. Examination by pathologists did not reveal tumors or premalignant lesions such as dysplasia or adenomas, even after 8 months after the doxycycline pulse. Non-neoplastic pathological changes of kidney tissues like cysts, hypertrophy, necrosis, dilation or atrophy of tubules as well as tubule-interstitial fibrosis were neither present (Fig 6E). Nuclear crowding was observed in some tubules in the cortex in β-catenin-GOF; Notch-GOF, but not in Vhl-LOF; β-catenin-GOF kidneys (Fig 6F). We conducted Ki67 staining for proliferating cells at the G1, S, G2 and M phases of the cell cycle. None of mutant kidneys, even after 8 months after the doxycycline pulse, exhibited higher numbers of Ki67-positive nuclei than controls (Fig 6G and 6H), despite upregulation of the genes of Cyclin D1 and Myc involved in cell cycle progression.

We neither detected accumulation of phospho-γH2AX nor p53 stabilization and the activation of its downstream target p21 in nuclei of mutant kidney cells (S6A Fig). We also did not observe increases in the numbers of apoptotic cells as examined by TUNEL assays (S6B Fig). Together, we neither found signs of malignant transformation nor of DNA damage, senescence and cell death in mutant kidneys.

## Mutant mice displayed phenotypes of CKD

To confirm anemia in mutant mice, we determined the hematocrit, which significantly decreased in mutant mice (Fig 7A). We also measured the concentration of erythropoietin (EPO) in the blood plasma, which strongly rose in mutant mice (Fig 7B). Pathological

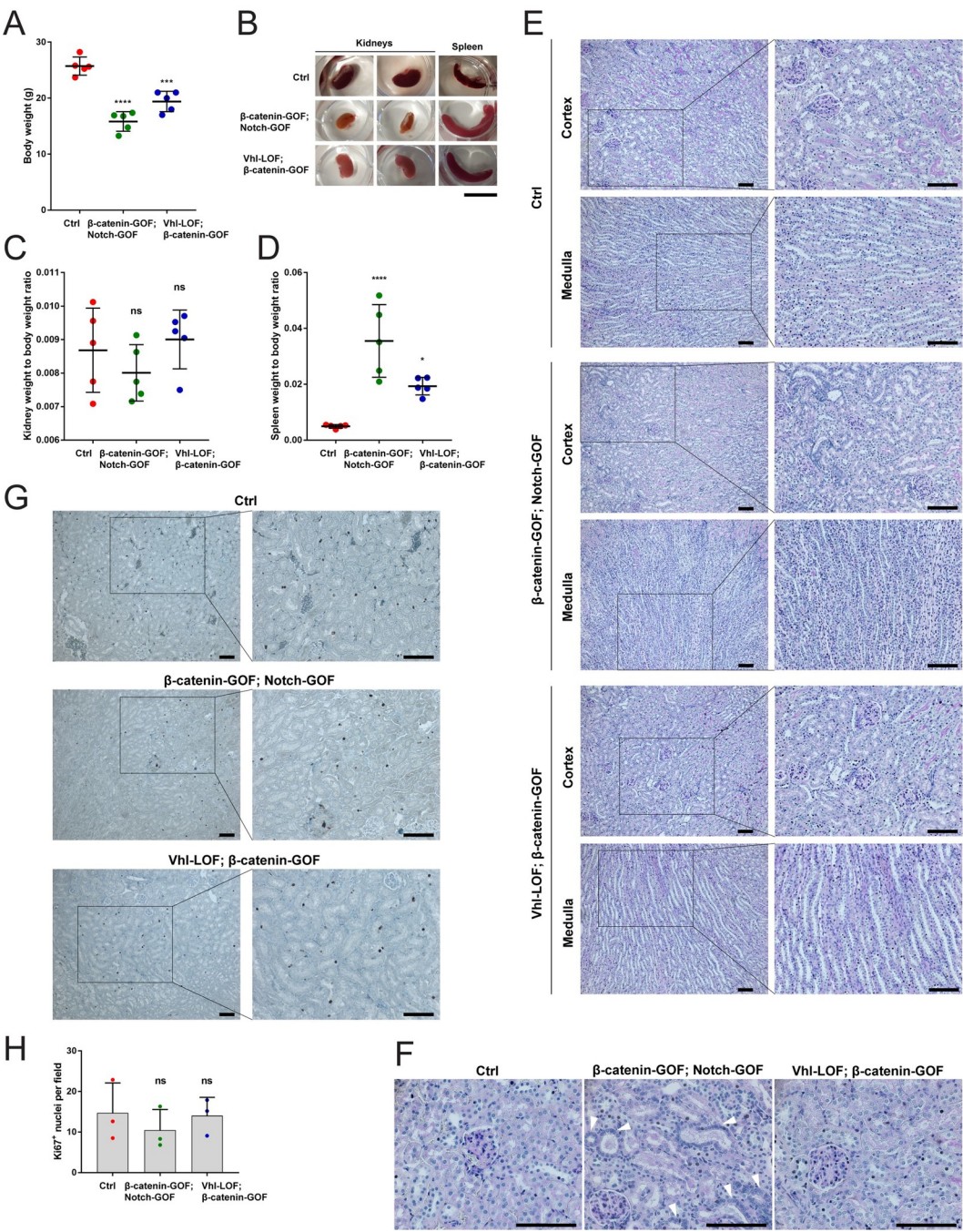

**Fig 6. No tumorigenesis was observed in double mutant kidneys.** (A) Body weight of β-catenin-GOF; Notch-GOF and Vhl-LOF; β-catenin-GOF mutant mice versus controls. (B) Overview of kidneys and spleens of β-catenin-GOF; Notch-GOF and Vhl-LOF; β-catenin-GOF mutant mice versus controls. (C) Kidney weight to body weight ratio of β-catenin-GOF; Notch-GOF and Vhl-LOF; β-catenin-GOF mutant mice versus controls. (D) Spleen weight to body weight ratio of β-catenin-GOF; Notch-GOF and Vhl-LOF; β-catenin-GOF mutant mice versus controls. (E) PAS staining in β-catenin-GOF; Notch-GOF and Vhl-LOF; β-catenin-GOF mutant kidneys versus controls (in the cortex and medulla) after 8 months after the doxycycline pulse. (F) PAS-stained outer part of the cortex with nuclear crowding (arrowheads) in some tubules in β-catenin-GOF; Notch-GOF, but not in Vhl-LOF; β-catenin-GOF mutant kidneys. (G) Immunohistochemistry for nuclear Ki67 in β-catenin-GOF; Notch-GOF and Vhl-LOF; β-catenin-GOF mutant cortical kidneys versus controls after 8 months after the doxycycline pulse. (H) Quantification of Ki67-positive nuclei in β-catenin-GOF; Notch-GOF and Vhl-LOF; β-catenin-GOF mutant cortical kidneys versus controls after 8 months after the doxycycline pulse. Data information: in A, C and D, 5 independent replicates (mice, dots) per line were examined. Graphs show mean ± SD (error bars). Data passed the Shapiro-Wilk normality test (α = 0.05). The ordinary one-way ANOVA followed by the Dunnett's multiple comparison to

controls was performed; P-values from left to right in A; 0.0001, 0.0002; ****p < 0.0001, ***p < 0.001; in C; 0.4879, 0.8354; ns: non-significant; in D; 0.0001, 0,0229; ****p < 0.0001, *p < 0.05. Scale bar in B, 1 cm. 20 independent replicates (all mice induced) per line were examined. Scale bars in E-G, 100 μm. Nuclei are counterstained with haematoxylin. In E and G, insets are enlarged on the right. 20 independent replicates (all mice induced) per line were examined. In H, three independent replicates (mice, dots) with 10 technical replicates (fields per kidney) each per line were examined. Graphs show mean + SD (error bars). Data passed the Shapiro-Wilk normality test (α = 0.05). The ordinary one-way ANOVA followed by the Dunnett's multiple comparison to controls was performed; P-values from left to right, 0.6019, 0.9871; ns: non-significant.

examination of the enlarged spleens of mutant mice revealed extramedullary hematopoiesis by the presence of megakaryocytes (Fig 7C).

We found upregulation of tubular injury markers *Ngal*, *Kim-1* and *Gsta1* in mutant kidneys (Fig 7D). Moreover, the concentration of another marker of kidney injury, blood urea nitrogen (BUN), was increased in mutant mice (Fig 7E). We also examined in mutant kidneys expression of the inflammatory cytokine *Il1b* and of a panel of markers for myeloid lineage immune cells, *Cd11b*, *Ly6c*, *Ccr2* and *Cd68*, as well as of *Icam1*, a ligand for *Cd11b*. Expression of *Il1b* and *Cd11b* was upregulated in both kidney mutants, while expression of *Ly6c*, *Ccr2*, *Cd68* and *Icam1* was increased in Vhl-LOF; β-catenin-GOF mutant, but not in β-catenin-GOF; Notch-GOF mutant (Fig 7F). Expression of fibrotic markers, i.e. markers of activated myofibroblasts, *a-sma* and *Vim*, and markers of extracellular matrix, *Fn1*, *Col1a1* and *Col3a1*, was not enhanced in mutant kidneys (Fig 7G). We conclude that mutant mice exhibited lethal features of CKD such as anemia as well as kidney injury and kidney inflammation. Thus, we decided not to proceed with the generation of Vhl-LOF; β-catenin-GOF; Notch-GOF triple mutant mice.

## Discussion

We have established robust tubuloid cultures from single epithelial cells from adult mouse kidneys and have examined their cellular components and stem cell-associated signaling systems at protein levels by deep bulk proteomic and phosphoproteomic analyses. These approaches represent so far the most thorough recording of important proteins and phosphorylation sites performed in human and mouse adult kidney organoids/tubuloids. We found that tubuloids shared proteome and phosphoprotome patterns with adult kidney tubular epithelia to a great extent and were phenotypically stable over long-term passages. Proteomic analysis revealed enhanced proliferation and upregulation of stem cell-associated Wnt, Notch and Yap signaling in tubuloids, which indicated the recapitulation of kidney renewal response, similar to mice upon kidney injury [3, 9–11, 46]. Functionally, Wnt was crucial for both expansion and differentiation of tubuloid-forming cells. Despite high coverage, we were not able to detect important proteins such as collecting duct-specific Aquaporin 3, which we detected using other techniques. We did not examine the functional importance of phosphorylation patterns of detected proteins, because this went beyond the scope of our manuscript. However, we believe that the phosphoproteomic data set holds a lot of interesting information for other researchers regarding signaling system in adult kidney cells with stem/progenitor-like functionality.

To characterize tubuloids more thoroughly, single cell RNA sequencing should be performed in the future. Once optimized, single cell proteomics will remarkably improve a global analysis of protein levels in individual tubuloid cells [47]. In a very recent study on iPSC-derived kidney organoids, proteome trajectories over culture duration vere defined and integrated with single cell transcriptomic profiles [48]. As bulk proteomics does not specify which cells displayed enhanced Wnt and Notch signaling, correlating single cell data on Wnt and Notch upregulation and expression of specific markers (surface proteins or transcriptional factors) will be highly important as a first step to identify cell populations with stem/progenitor-

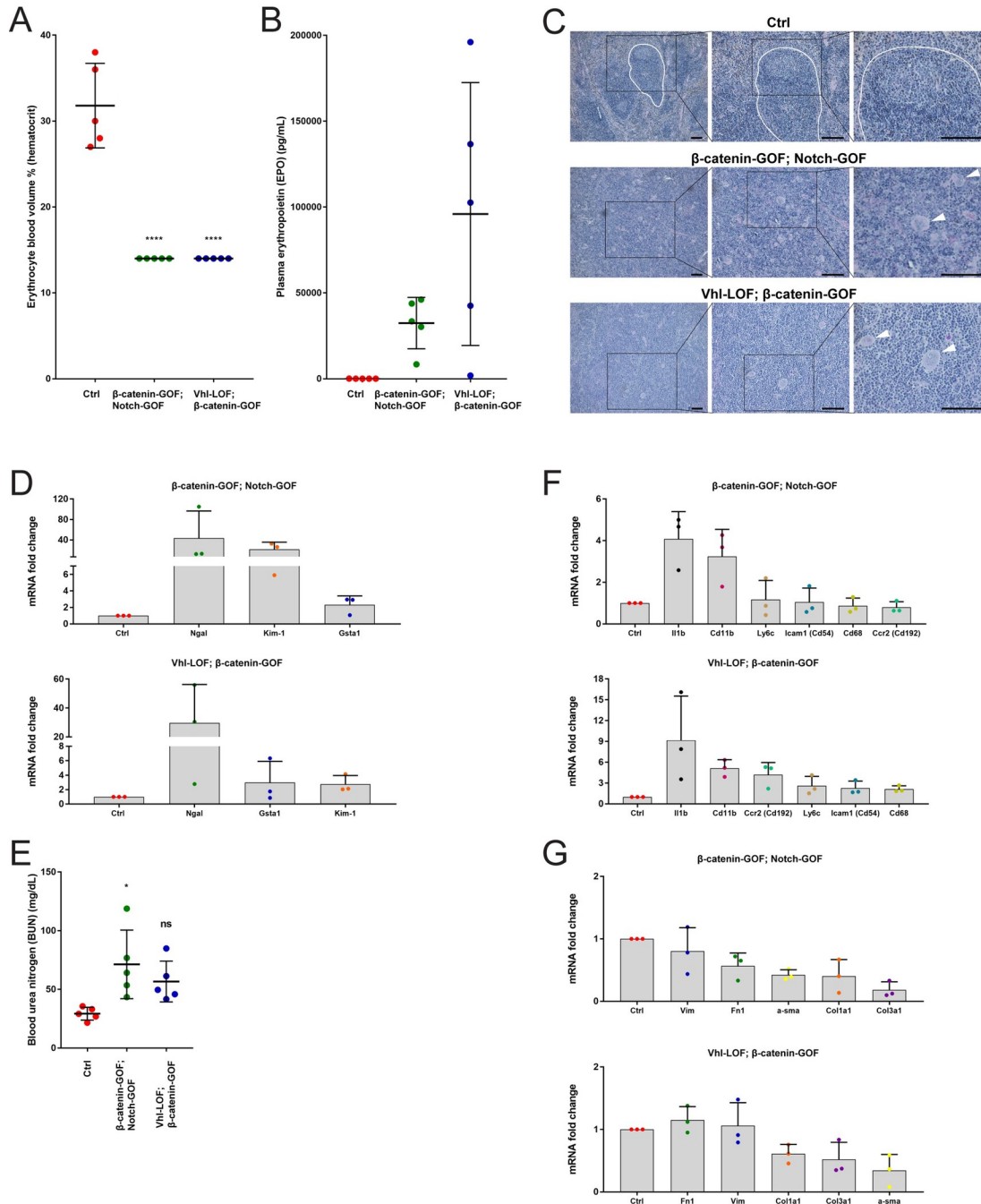

**Fig 7. Mutant mice displayed phenotypes of chronic kidney disease (CKD).** (A) Hematocrit of β-catenin-GOF; Notch-GOF and Vhl-LOF; β-catenin-GOF mutant mice versus controls. (B) Concentration of plasma EPO in β-catenin-GOF; Notch-GOF and Vhl-LOF; β-catenin-GOF mutant mice versus controls. (C) PAS staining showing extramedullary hematopoiesis in spleens of β-catenin-GOF; Notch-GOF and Vhl-LOF; β-catenin-GOF mutant mice versus controls. Megakaryocytes are marked by arrowheads. (D) Fold change of gene expression of kidney injury markers in β-catenin-GOF; Notch-GOF and Vhl-LOF; β-catenin-GOF mutant kidneys versus controls. (E) Concentration of blood urea nitrogen (BUN) in β-catenin-GOF; Notch-GOF and Vhl-LOF; β-catenin-GOF mutant mice versus controls. (F) Fold change of gene expression of inflammation markers in β-catenin-GOF; Notch-GOF and Vhl-LOF; β-catenin-GOF mutant kidneys versus controls. (G) Fold change of gene expression of fibrotic markers of myofibroblasts and extracellular matrix in β-catenin-GOF; Notch-GOF and Vhl-LOF; β-catenin-GOF mutant kidneys versus controls. Data information: in A, 5 independent replicates (mice, dots) with one technical replicate each per line were examined. Graphs show mean ± SD (error bars). Data passed the Shapiro-Wilk normality test (α = 0.05). The ordinary one-way ANOVA followed by the Dunnett's multiple comparison to controls was performed; P-values from left to right, 0.0001, 0.0001; ****p < 0.0001. In B, 5 independent replicates (mice, dots) in technical triplicates per line were examined. Graphs show

mean ± SD (error bars). No statistical test was performed because of high inter-mouse variance. However, increase trends were observed. Scale bars in C, 100 μm. Nuclei are counterstained with haematoxylin. An undisrupted follicular B-cell compartment in the white pulp in controls is marked by white lines. Insets are enlarged on the right. Three independent replicates per line were examined. In D, three independent replicates (mice, dots) in technical triplicates were examined. Graphs show mean + SD (error bars). No statistical test was performed because of high inter-mouse variance. However, increase trends were observed. In E, 5 independent replicates (mice, dots) with one technical replicate each per line were examined. Graphs show mean ± SD (error bars). Data passed the Shapiro-Wilk normality test (α = 0.05). The ordinary one-way ANOVA followed by the Dunnett's multiple comparison to controls was performed; P-values from left to right, 0.0109, 0.0888; *p < 0.05, ns: non-significant. Although the difference in average concentration of BUN between the Vhl-LOF; β-catenin-GOF mutant mice and controls was non-significant (p < 0.05), an increase trend was observed. In F and G, three independent replicates (mice, dots) in technical triplicates were examined. Graphs show mean + SD (error bars). No statistical test was performed because of high inter-mouse variance. However, increase trends were observed in F and no increase trends were observed in G.

like functionality that drive tubuloids as well as to determine their origin from resident self-renewing cells or de-differentiated mature cells in tubular epithelia. It might be then possible to link these cells with their malignant counterparts, for instance, with the ones identified by our group [19], and to study Wnt- and Notch-driven tumorigenesis *in vitro*. Together, tubuloid cultures from adult mouse kidneys enrich for Wnt- and Notch-high cells with stem/progenitor-like functionality.

Surprisingly, the two types of double kidney mutants in mice, i.e. β-cat-GOF with Notch-GOF or with Vhl-LOF, did not produce tumors or premalignant lesions. This occurred despite a rational hypothesis on the involvement of Wnt- and Notch-high stem cells in development of human ccRCC, and despite using a well-designed genetic system to target adult proximal tubular epithelial cells, the cells of tumor origin. However, the mouse mutants displayed strong fatal phenotypes that resembled CKD [49, 50]. The significance of the interplay of Wnt and Notch as well as of Wnt and Vhl loss in development of CKD has not previously been defined.

Dysregulation of Wnt and Notch is observed in patients with different forms of tubular and glomerular CKD [51]. Wnt hyperactivation using embryonic Cres in other mouse models resulted in development of polycystic kidney disease [52, 53]. Wnt upregulation in tubular cells or podocytes of adult mouse kidneys contributed to either tubular and glomerular damage and tubule-interstitial inflammation or to albuminuria, indicating CKD [54, 55]. In turn, adult kidneys with Notch hyperactivation exhibited CKD with pronounced tubular degeneration and dilation as well as tubule-interstitial inflammation and fibrosis [56]. By single cell RNA sequencing, plasticity of collecting duct cells in the mouse kidney was revealed, which upon Notch upregulation was disturbed and correlated with metabolic acidosis observed in CKD patients [57]. Moreover, in numerous rodent models of kidney injury, transient activation of Wnt and Notch accelerated regeneration, but sustained activation promoted maladaptative behaviors and CKD progression [51, 58]. Of note, Notch upregulation in mouse kidneys in another study led to development of both CKD and a relatively rare papillary kidney cancer, and this process was accelerated by kidney injury [59]. In addition, the authors found a correlation of a kidney injury episode with subsequent occurrence of papillary cancer but not ccRCC in patients.

Lack of fibrosis in our models might suggest that mutant mice have not yet progressed to end-stage CKD. Remarkably, mutant mice produced additional features of CKD, which so far have not been reported in other studies that examined hyperactivation of either Wnt or Notch in adult mouse kidneys [55, 56]. Mutant mice exhibited dramatic anemic phenotypes, despite strong increases in plasma erythropoietin levels. This compensatory positive feedback on EPO production suggests a mechanism of anemia development downstream of EPO supply. Inflammation that we detected in mutant mice could be a driver of anemia in CKD, either directly through inhibition of EPO-mediated erythropoiesis or indirectly via disrupted iron regulation that limits hemoglobin synthesis in erythroid cells in the bone marrow [50]. Moreover, we

documented extramedullary hematopoiesis in enlarged spleens of mutant mice, which may be induced in response to anemia and involves expansion and differentiation of erythroid, myeloid and platelet precursors into effector cells outside the bone marrow [60].

We suspect that focal recombination in adult kidney epithelial cells in mice through decreased doses of doxycycline might support tumor development by overcoming severe CKD phenotypes [61, 62]. Thus, our reproducible and penetrant genetic system can be used for further studies of downstream effectors of Wnt and Notch signaling in the context of the pathobiology of kidney stem/progenitor cells, irrespective whether this results in CKD or tumors. We conclude that the tubuloid model displayed responses of adult kidney cells with stem/progenitor-like functionality that resulted in high self-renewal, in contrast to the genetic mouse mutants that produced CKD phenotypes, but no elevated proliferation and tumors. In line with previous studies in mice, tightly regulated endogenous Wnt and Notch signaling drove regeneration, but contributed to CKD when genetically dysregulated. Together, our findings in genetic models of adult mouse kidneys challenge the current understanding of the involvement of stem cell-associated Wnt and Notch signaling in development of human ccRCC.

## Supporting information

**S1 Fig. Long-term culture of kidney tubuloids and establishing kidney tubuloids of different mouse backgrounds.**
(PDF)

**S2 Fig. Yap signaling is upregulated in kidney tubuloids.**
(PDF)

**S3 Fig. Expression of selected markers of resident stem/progenitor cells in tubuloids.**
(PDF)

**S4 Fig. Markers of tubular epithelial cells are expressed in 3D-reconstructed kidney tubuloids.**
(PDF)

**S5 Fig. Kidney tubuloids display tubular epithelial polarity and complexity.**
(PDF)

**S6 Fig. No DNA damage, growth arrest or senescence and increased apoptosis was observed in mutant kidneys.**
(PDF)

**S1 Video. Markers of tubular epithelial cells are expressed in 3D-reconstructed kidney tubuloids.**
(MP4)

## Acknowledgments

We would like to thank Prof. Dr. med. Wolfgang Schneider and Dr. med. Ann-Christin von Brünneck from the Institute of Pathology at the Charité-Medical University Berlin for the histopathological examinations. Further, we thank Prof. Dr. med. Friedrich Luft from the Experimental and Clinical Research Center (ECRC) in Berlin for giving us important experimental suggestions. We would also like to acknowledge Dr. Bart Spee from the Faculty of Veterinary Medicine at the Utrecht University for his valuable comments on the manuscript.

## Author Contributions

**Conceptualization:** Adam Myszczyszyn, Annika Fendler, Walter Birchmeier.

**Data curation:** Adam Myszczyszyn, Oliver Popp, Severine Kunz, Anje Sporbert, Simone Jung.

**Formal analysis:** Adam Myszczyszyn, Oliver Popp, Severine Kunz, Anje Sporbert.

**Funding acquisition:** Walter Birchmeier.

**Investigation:** Adam Myszczyszyn, Oliver Popp, Severine Kunz, Simone Jung.

**Methodology:** Adam Myszczyszyn, Oliver Popp, Severine Kunz, Anje Sporbert, Annika Fendler, Philipp Mertins, Walter Birchmeier.

**Resources:** Oliver Popp, Severine Kunz, Anje Sporbert, Philipp Mertins.

**Software:** Oliver Popp, Severine Kunz, Anje Sporbert, Philipp Mertins.

**Supervision:** Philipp Mertins, Walter Birchmeier.

**Validation:** Adam Myszczyszyn, Oliver Popp, Severine Kunz, Anje Sporbert, Philipp Mertins.

**Visualization:** Adam Myszczyszyn, Oliver Popp, Severine Kunz, Anje Sporbert.

**Writing – original draft:** Adam Myszczyszyn, Louis C. Penning, Walter Birchmeier.

**Writing – review & editing:** Adam Myszczyszyn, Louis C. Penning, Walter Birchmeier.

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
