## [Decision Letter · Decision Letter 0]

11 Jul 2023

PONE-D-23-05670Mice with renal-specific alterations of stem cell-associated signaling develop symptoms of chronic kidney disease but surprisingly no tumorsPLOS ONE

Dear Dr. Myszczyszyn,

Thank you for submitting your manuscript to PLOS ONE. After careful consideration, we feel that it has merit but does not fully meet PLOS ONE’s publication criteria as it currently stands. Therefore, we invite you to submit a revised version of the manuscript that addresses the points raised during the review process. Please respond to all reviewers' comments, particularly those of reviewer #2, which are extensive.  Reviewer #1 suggests changing the terminology for the spherical cellular aggregates (organoids).  I think you can better diffuse how the structures formed in your work differ from the current common usage of the term orgaoids.  

We look forward to receiving your revised manuscript.

Kind regards,

Michael Klymkowsky, Ph.D.

Academic Editor

PLOS ONE

Journal Requirements:

3. Please make sure that all information entered in the 'Ethics Statement' section regarding ethics approval is also included in the Methods section of the manuscript.

A.M. and A.F. were funded in part by the Urological Research Foundation (Stiftung Urologische Forschung) in Berlin.

Reviewers' comments:

Reviewer's Responses to Questions

**Comments to the Author**

1. Is the manuscript technically sound, and do the data support the conclusions?

Reviewer #1: Yes

Reviewer #2: Partly

2. Has the statistical analysis been performed appropriately and rigorously? 

Reviewer #1: Yes

Reviewer #2: Yes

3. Have the authors made all data underlying the findings in their manuscript fully available?

Reviewer #1: Yes

Reviewer #2: Yes

4. Is the manuscript presented in an intelligible fashion and written in standard English?

Reviewer #1: Yes

Reviewer #2: Yes

5. Review Comments to the Author

Reviewer #1: In this study, the authors performed proteomic and phosphoproteomic analyses on spheres generated from dissociated kidney cells, and modulated WNT or NOTCH pathways in the spheres. The authors also generated transgenic mice with β-catenin-GOF; Notch-GOF and Vhl-LOF; β-catenin-GOF, and observed no tumorigenesis in these mutant kidneys.

The manuscript is informative that β-catenin-GOF; Notch-GOF cannot cause tumorigenesis in mice in vivo. Because it is not convincing that the spheres faithfully recapitulate endogenous biological processes, the data from the sphere experiments are not so helpful.

Prior to publication in PLOS ONE, the following points must be addressed.

Major point:

1. To avoid unnecessary confusion in readers, "organoid" should be replaced with "spheres" throughout the manuscript. Organoids indicate miniature organs recapitulating the key functional, structural and biological complexity of the organ. The authors merely generated spheres from dissociated kidney cells, which are not organoids, not kidney organoids containing multiple kidney cell types often generated from pluripotent stem cells.

Minor points:

2. Due to the heterogenous spheres of different cell types, it is unclear spheres of which cell types were affected by modulation of WNT or NOTCH pathways.

3. It is very unclear which transgene caused CKD in mice. Is β-catenin-GOF is sufficient for CKD? Does Notch-GOF or Vhl-LOF play any role in CKD?

Reviewer #2: This is an interesting study in which to reveal the importance of Wnt and Notch signaling for renal cells in vitro, without enriching populations positive for a specific marker, the authors have established tubular organoids from whole mouse kidneys. To mimic development of human ccRCC from Wnt- and Notch-dependent cells, they generated genetic mutants in mice, showing an important role of these signaling for CKD.

However, there are important issues to address.

1. The authors reported that they isolated single epithelial cells from whole adult mouse kidneys and established 3D organoid cultures but in the paper they speak about renal stem/progenitor cells. They used freshly isolated cell suspensions derived from whole kidneys without any sorting, therefore we cannot be sure that generated organoids were derived by stem/progenitor cells. The authors must therefore speak about organoids derived from bulk adult kidney epithelial cells

2. Have the authors tried to generate organoids from Epcam-positive kidney epithelial cells ? This control is needed to evaluate the origin of organoids.

3. What the authors mean for Self-renewal of organoids? It’s hard indicate self-renewal capacity of cells only from the observation that organoids formation from single cells was five times higher in the first passage in comparison to freshly seeded cultures. I would not self-renewal at all.

4. In proteome analysis what are the differences between cells derived from organoids and EPCAM+ cells? This is not clear and it is very important also for the remaining part of the study regarding RCC.

5. The authors asserted that Kidney organoids displayed enhanced proliferation. Where is the comparison between BrdU cells derived from organoids and EPCAM+ and adult kidney epithelial cells (control kidney)? All proliferating cells incorporate BrdU, therefore Figure 2D is not sufficient to assert that Kidney organoids displayed enhanced proliferation.

6. The observation that Wnt signaling controlled growth and differentiation of kidney organoids is very interesting. Why R-spondin-1 removal led to a switch from predominant cystic to fully solid organoids, while treatment with ICG-001 at IC50 did not result in morphological changes? The authors could try to provide and explanation and this should be discussed

7. Sentence in lines 613-618 are not clear. Please revise.

8. In the discussion I suggest to remove paragraph headings in bold. The first heading “Tubular organoid cultures from adult mouse kidneys enrich for stem/progenitor cells with active Wnt and Notch signaling”, as well as the second one, are not corrected for the reason explained in point 1.

9. I suggest to focus the entire paper on the important role of Wnt and Notch that transiently activated can accelerate the regeneration but sustained activated can promove maladaptative behaviors and CKD progression. This is a very interesting point.

10. The role of WNT and notch signaling in normal human adult renal stem/progenitor cells that has been reported in several studies (Please see PMID: PMID: 35922038, PMID: 37190024, PMID: 23861881, PMID: 19843711) should be also discussed.

6. PLOS authors have the option to publish the peer review history of their article (what does this mean?). If published, this will include your full peer review and any attached files.

Reviewer #1: No

Reviewer #2: No

---

## [Author Response · Author response to Decision Letter 0]

24 Aug 2023

Berlin, August 24th, 2023

Rebuttal letter

Dear dr. Klymkowsky,

We would like to thank you and both reviewers for the efforts in assessing our manuscript. Our impression is that you appreciate merit of the organoid and genetic system and that you find the study attractive for readers of PLOS ONE. This is especially important for us, because we have aimed to provide the kidney research community with models that can serve as a tool to further examine normal and disease renal cells with stem/progenitor-like functionality. Within the given time of 45 days, we have been working hard to prepare thorough responses to the individual points of criticism. We hope that these will satisfy you and reviewers, and will help readers in understanding the findings of the study, as we agree to publish the revision report along with the manuscript. 

We have now adjusted the manuscript to the style requirements of PLOS ONE. We have also provided additional information on the animal experiments in Materials and Methods: The Landesamt für Gesundheit und Soziales (LaGeSo) in Berlin approved the mouse study (G0342/13). Animal experiments were conducted in accordance with European, national, federal state and institutional regulations. To alleviate suffering, mice were checked at least twice per week according to a score sheet until illness symptoms were observed. Sick mice were immediately anesthesized with isoflurane and sacrificed by cervical dislocation, then biological material was collected. We have updated the Ethics Statement accordingly. Furthermore, we have removed the funding-related text from the manuscript. We would still like to acknowledge the Urological Research Foundation (Stiftung Urologische Forschung) in Berlin for funding a doctoral position of A.M. and a postdoctoral position of A.F. However, the Foundation had no role in study design, data collection and analysis, decision to publish, or preparation of the manuscript. We believe that we can find a compromise. Please then, adjust the Funding Statement in the online submission form on our behalf. Finally, once the manuscript is accepted, we will make the data deposited to the ProteomeXchange Consortium (PXD023491) publically accessible. To access the data now, please use as username reviewer_pxd023491@ebi.ac.uk and as password rYMEiE9t. 

We look forward to hearing from you and would be glad to respond to any further questions or comments.

Yours sincerely,

Dr. Adam Myszczyszyn, first author

Prof. Dr. Walter Birchmeier, senior author

Reviewer #1: 

In this study, the authors performed proteomic and phosphoproteomic analyses on spheres generated from dissociated kidney cells, and modulated WNT or NOTCH pathways in the spheres. The authors also generated transgenic mice with β-catenin-GOF; Notch-GOF and Vhl-LOF; β-catenin-GOF, and observed no tumorigenesis in these mutant kidneys. 

This is an accurate description of our study, apart from the modulation of Wnt and Notch. Upregulation of these signaling systems in tubular organoids (tubuloids) was endogenous. We blocked Wnt with small-molecule inhibitors that act downstream of the pathway. 

The manuscript is informative that β-catenin-GOF; Notch-GOF cannot cause tumorigenesis in mice in vivo. Because it is not convincing that the spheres faithfully recapitulate endogenous biological processes, the data from the sphere experiments are not so helpful. 

Thank you for appreciating our genetic model. We are sorry that you find our in vitro model not fully informative. We agree that tubuloids do not resemble all the nephron segments and non-epithelial cell types. For this, excellent iPSC-derived organoid models were established in the last years [1]. We argue that our mouse tubuloid model partly recapitulates renewal responses of adult kidney epithelia, similar to human kidney tubuloids and epithelial organoids from other organs that were generated in the group of a worldwide expert in the field, Hans Clevers [2,3]. This is not possible with iPSC-derived organoids, as these resemble the fetal kidney [1]. In this light, the tubuloid model is useful in interpreting the in vivo data.

Prior to publication in PLOS ONE, the following points must be addressed:

Major point:

1. To avoid unnecessary confusion in readers, "organoid" should be replaced with "spheres" throughout the manuscript. Organoids indicate miniature organs recapitulating the key functional, structural and biological complexity of the organ. The authors merely generated spheres from dissociated kidney cells, which are not organoids, not kidney organoids containing multiple kidney cell types often generated from pluripotent stem cells.

You are correct that purely epithelial Matrigel-based 3D organoids derived from adult stem cells do not reach the biological complexity of iPSC-derived organoids. Though, we argue that the term spheres would confuse readers even more. Spheres are just cellular aggregates that can grow in floating conditions; see our previous study, for instance [4]. In contrast, we generated 3D structures grown from single cells that self-organize in ECM-mimicking Matrigel and serum-free medium with defined growth factors (Fig 1A), form a lumen and are polarized (Fig 3D), and thus resemble classical cystic adult stem cell-derived organoids from numerous epithelial organs that were established in the group of Hans Clevers since 2009 [3]. Although a term spheroids is sometimes used to describe cystic organoids, these 3D structures in other studies cannot be cultured in a long-term fashion [5] like our ones (S1A Fig). For a compromise, we have now changed the term organoids to tubuloids (tubular organoids), as we grew these from murine tubular epithelial cells. This is in line with human tubuloids generated in the Clevers’ group [2].

Minor points:

2. Due to the heterogenous spheres of different cell types, it is unclear spheres of which cell types were affected by modulation of WNT or NOTCH pathways.

We apologize for this apparent unclarity, but we did not modulate Wnt and Notch genetically in tubuloids. Upregulation of these signaling systems was endogenous. We blocked Wnt with small-molecule inhibitors that act downstream of the pathway. We agree that bulk proteomics does not specify which cells displayed enhanced Wnt and Notch signaling. For this, single cell RNA (or ideally protein) sequencing will be useful. Correlating single cell data on Wnt and Notch upregulation and expression of specific markers (surface proteins or transcriptional factors) will be highly important as a first step to identify cell populations with stem/progenitor-like functionality that drive tubuloids as well as to determine their origin from resident self-renewing cells or de-differentiated mature cells in tubular epithelia. We have now updated Discussion accordingly, and have also commented on a very recent study on iPSC-derived kidney organoids that defined proteome trajectories over culture duration and integrated those with single cell transcriptomic profiles [6] (see page 34 and 35, Revised Manuscript with Track Changes).

3. It is very unclear which transgene caused CKD in mice. Is β-catenin-GOF is sufficient for CKD? Does Notch-GOF or Vhl-LOF play any role in CKD?

We focused on double mutants, especially β-catenin-GOF; Notch-GOF, as we aimed to mimic development of human clear cell kidney tumors from Wnt- and Notch-high cancer stem cells that we identified previously [4]. β-catenin-GOF was sufficient for inducing chronic kidney disease-like phenotypes in two only single mutant mice that we generated (data not shown). Most probably, it was also a driver of the phenotypes in double mutants with Vhl-LOF, as Vhl does not play any role in the development of chronic kidney disease, to our best knowledge. This role of β-catenin-GOF as a single driver of chronic kidney disease is supported by previous studies in mouse models. Also, Notch-GOF alone was shown to induce chronic kidney disease in mice. We believe that this knowledge is already sufficiently elaborated in the manuscript (see Discussion, page 36, Revised Manuscript with Track Changes). 

Reviewer #2: 

This is an interesting study in which to reveal the importance of Wnt and Notch signaling for renal cells in vitro, without enriching populations positive for a specific marker, the authors have established tubular organoids from whole mouse kidneys. To mimic development of human ccRCC from Wnt- and Notch-dependent cells, they generated genetic mutants in mice, showing an important role of these signaling for CKD. 

This is an excellent summary of our study. Thank you.

However, there are important issues to address:

1. The authors reported that they isolated single epithelial cells from whole adult mouse kidneys and established 3D organoid cultures but in the paper they speak about renal stem/progenitor cells. They used freshly isolated cell suspensions derived from whole kidneys without any sorting, therefore we cannot be sure that generated organoids were derived by stem/progenitor cells. The authors must therefore speak about organoids derived from bulk adult kidney epithelial cells.

This is a relevant comment and we agree that our findings on the tubuloid system would be more informative if we identified individual stem/progenitor cell populations based on surface markers or transcription factors. We have now examined expression of three markers associated with human and mouse resident renal stem/progenitor cells, i.e. Prom1 (Cd133) [7–9], Sox9 [8,10] and Pax2 [7]. However, Prom1 was downregulated in tubuloids, as compared to freshly isolated Epcam-positive kidney epithelial cells (Revision Fig 1A). Provocatively, a lineage tracing of Prom1-positive cells in the mouse kidney questioned their stem/progenitor cell functionality, as these cells displayed limited generative capacity in the postnatal kidney and were quiescent in the adult kidney. Neither, upregulation of either Wnt or Notch in these cells in the adult kidney produced tumors [11]. We detected Sox9 neither in tubuloids nor in freshly isolated Epcam-positive kidney epithelial cells due to either a technical issue related to coverage or a very low expression. Alternatively, we did not find the presence of Sox9 in tubuloids using immunohistochemistry (Revision Fig 1B). Moreover, a slight downregulation of Pax2 in tubuloids was found, as compared to kidney cells (Revision Fig 1A). Nevertheless, tubuloid-forming efficiency at passage 0 in our system was only 0.04% (40 tubuloids on average grown from 105 freshly seeded epithelial cells, data not shown). This might be in favor of the presence of resident stem/progenitor cell populations, as the numbers of adult stem cells in their niches in vivo are very low [12]. 

Surprisingly, proteomic analysis revealed in tubuloids upregulation of Six2, a marker for an embryonic population of nephron progenitor cells [13] (Revision Fig 1A). This might suggest that tubuloid-forming cells are a result of a de-differentiation process, but further studies to confirm a functional role of Six2-positive cells in tubuloids are necessary. Using proteomics, we also detected in tubuloids increased levels of Met and Cd44, which are the markers of cancer stem cells that we identified in human clear cell kidney tumors [4]. Both are also Wnt targets (Fig 2E). 

In contrast to sorted phenotypically defined populations of resident stem/progenitor cells, a refined definition of functional stemness was proposed by Hans Clevers [14]. It encompasses the ability to replace lost tissue through cell division in adult organs with a little turnover during homeostatic maintenance. The adult kidney is considered such a quiescent organ [8], and it was demonstrated that mature kidney proximal tubule epithelial cells can upregulate Prom1 and enter the cell cycle upon organ injury to contribute to repair [15]. Tubuloid-initiating cells in our system fulfill this stemness definition by displaying enhanced and long-term clonogenicity (Fig 1C and D and S1A Fig) and proliferation (Fig 2C and D), and by upregulating Wnt signaling (Fig 2E) that controls tubuloid-forming potential, expansion and differentiation (Figure 4). Two other stem/progenitor cell-associated signaling systems, Notch (Fig 2F) and Yap (S2 Fig), were also upregulated. Activity of Wnt and Notch seems to be a common feature of different cell populations in the kidney with stem/progenitor-like functionality [8,16–19]. 

Together, we cannot rule out a possibility that various cell populations with stem/progenitor-like functionality, resident or facultative, are enriched within our culture conditions. Similar to our protocol, most of the protocols for establishing organoid cultures from adult organs rely on activation of EGF and Wnt signaling, and inhibition of TGF-β signaling, which are crucial for driving self-renewal of various stem/progenitor-like cell populations [3]. We believe that in future studies, our tubuloid model will assist analyses of Wnt- and Notch-dependent marker-sorted populations of putative resident kidney stem/progenitor cells, which will then possibly link these cells with their malignant counterparts, for instance, with the ones identified by our group, and will enable studying Wnt- and Notch-driven tumorigenesis in vitro. 

To accommodate your valid point, we have now changed the term stem/progenitor cells to cells with stem/progenitor-like functionality in the manuscript. We have also included Revision Fig 1 as S3 Fig, together with a corresponding text in the Results (see page 22, Revised Manuscript with Track Changes).

Revision Fig 1. Expression of selected markers of resident stem/progenitor cells in tubuloids. (A) Proteomic heatmap for Six2, Prom1 and Pax2 in both early (OEP) and long-term (OLP) passage tubuloids in comparison to freshly isolated Epcam-positive kidney epithelial cells (control kidney, CK). (B) Immunohistochemistry for Sox9 in both early (OEP) and long-term (OLP) passage tubuloids. Data information: in A, the heatmap shows normalized log2 intensity values for three independent replicates of CK, OEP and OLP. A 5% FDR (adjusted P-value < 0.05) cutoff and a log2 fold change cutoff of > 0 were applied for both OEP over CK and OLP over CK. For Six2, OLP over CK values were not significant in a pairwise comparison. The values were scaled (z-score by row) with breaks from ≤ -2 to ≥ 2. Scale bars in B, 100 μm. A positive ctrl used, high-Wnt and high-Met mouse mammary gland tumors. Nuclei are counterstained with haematoxylin. One independent replicate per condition was examined.

2. Have the authors tried to generate organoids from Epcam-positive kidney epithelial cells? This control is needed to evaluate the origin of organoids.

In line with the study of the group of Hans Clevers on human tubuloids [2], we did not generate mouse tubuloids from Epcam-positive kidney epithelial cells. Nevertheless, we agree that this would serve as a most elegant comparison to freshly isolated Epcam-positive cells. Though, we argue that our approach is enough to evaluate the epithelial origin of tubuloids, as we detected similar expression levels of Epcam in tubuloids and uncultured Epcam-positive cells (data available via the ProteomeXchange Consortium, see Data Availability Statement). This means that sorting is not necessary, as nearly all freshly isolated kidney epithelial cells seem to express Epcam.

3. What the authors mean for self-renewal of organoids? It’s hard to indicate self-renewal capacity of cells only from the observation that organoids formation from single cells was five times higher in the first passage in comparison to freshly seeded cultures. I would not call it self-renewal at all.

We agree that additional functional analyses are necessary, for instance, a reporter assay in vitro [20], an implantation into a mouse kidney [7,9] or a lineage tracing in vivo [8] based on a specific surface marker to reliably examine self-renewal capacity of plated single tubuloid-initiating cells. Though, these are not possible without any information on surface markers. Thus, we have now changed the term self-renewal to tubuloid-initiating capacity that indicates self-renewal. This is in agreement with previous studies, which examined clonogenic efficiency of organoid cultures by counting the number of organoids grown from plated single cells (or clonogenic expansion of serially diluted single cells grown as 2D monolayers), and this efficiency informed about self-renewal potential of organoid-forming cells [8,9,20–22]. Indeed, according to another expert in the field, Melissa Little, organoid cultures are considered a valid system to study self-renewal ability of adult stem/progenitor cells in vitro, as stated: intestinal epithelial cells cultured in Matrigel could be passaged repeatedly even from single Lgr5-expressing stem cells, providing in vitro evidence for their long-term self-renewing potential [5]. As already presented in the manuscript, we were able to clonally expand our tubuloid cultures from single cells (Fig 1A, C, D) in a long-term fashion (S1A Fig).

4. In proteome analysis, what are the differences between cells derived from organoids and EPCAM+ cells? This is not clear and it is very important also for the remaining part of the study regarding RCC.

All the data presented in Figs 2C, E, F and 3C, F, G, and in S2 and S3A Figs demonstrate differentially expressed proteins between freshly isolated Epcam-positive kidney epithelial cells (downregulation mostly), and early and long-term passage tubuloids (upregulation mostly). Our focus was to examine renewal phenotypes of adult kidney tubular epithelia in vitro using tubuloids, in contrast to quiescent kidney cells during homeostatic maintenance, which is important in the context of the in vivo data, i.e. no enhanced proliferation and tumorigenesis but chronic kidney disease. 

In turn, to analyze using the DAVID bioinformatic tool (see Materials and Methods, Descriptive statistics and significance testing) biological processes driven by proteins that are upregulated in Epcam-positive kidney epithelial cells and downregulated in tubuloids, we have now subjected 9,000 proteins detected in the proteomic analysis to both the 0.1% FDR cutoff (adjusted P-value < 0.001) and log2 fold change cutoff of < -0.5 (> 0.5 before) for both early passage tubuloids over control kidney cells and long-term passage tubuloids over control kidney cells. 1345 proteins were selected. Using the entire mouse (Mus musculus) proteome as a background for the UNIPROT_ACCESSION identifier, 1242 proteins (DAVID IDs) were identified and analyzed. Defined DAVID defaults were used and the Enrichment Thresholds (EASE Scores, modified Fisher-exact P-values) and adjusted P-values (Benjamini-Hochberg correction) for all terms in each cluster were subjected to the cutoff of < 0.001. 34 clusters were determined, mostly involved in energy production, i.e. mitochondrial oxidative phosphorylation, tricarboxylic acid cycle and fatty acid metabolism. These processes are extremely pronounced in the kidney [23,24], possibly to protect the organ against the consequences of the ischemia-reperfusion injury [23]. 

In the context of kidney regeneration in vitro, we have now analyzed by manual curation expression of classical transmembrane transporters of different mature nephron segments in early and long-term passage tubuloids versus control kidney cells. Expression of Slc34a1 (proximal tubule), Slc12a1 (Loop of Henle) and Slc12a3 (distal tubule) was detected in early and long-term passage tubuloids on protein level, but was downregulated in comparison to control kidney cells (Revision Fig 2). Taking these points into consideration, we conclude that our tubuloids display upregulation of only some markers specific for proximal and distal kidney tubular epithelia, as documented in Fig 3. Thus, improving tubuloid differentiation is necessary in future studies.

Revision Fig 2. Proteomic heatmap for Slc12a3, Slc34a1 and Slc12a1 in both early (OEP) and long-term (OLP) passage tubuloids in comparison to freshly isolated Epcam-positive kidney epithelial cells (control kidney, CK). Data information: the heatmap shows normalized log2 intensity values for three independent replicates of CK, OEP and OLP. A 5% FDR (adjusted P-value < 0.05) cutoff and a log2 fold change cutoff of > 0 were applied for both OEP over CK and OLP over CK. The values were scaled (z-score by row) with breaks from ≤ -2 to ≥ 2.

5. The authors asserted that Kidney organoids displayed enhanced proliferation. Where is the comparison between BrdU cells derived from organoids and EPCAM+ and adult kidney epithelial cells (control kidney)? All proliferating cells incorporate BrdU, therefore Figure 2D is not sufficient to assert that Kidney organoids displayed enhanced proliferation.

Our conclusion is based on the data presented in Fig 2C. We demonstrate the top three processes that were governed by the upregulated proteins in early and long-term passage tubuloids versus freshly isolated Epcam-positive kidney epithelial cells, i.e. cell cycle, ribosomal activity and DNA replication, which indicated the presence of proliferating cells over a long-term culture. For instance, the typical proliferation markers Ki67, Pcna and Cyclin D1 were upregulated in tubuloids versus control kidney cells (data available via the ProteomeXchange Consortium, see Data Availability Statement; for Cyclin D1, see also Fig 2E; we have now added this to the Results, see page 21, Revised Manuscript with Track Changes). BrdU incorporation in Fig 2D served as a functional confirmation of a high number of proliferating cells in tubuloids. We did not aim to compare BrdU incorporation levels between tubuloids and control kidney cells, as this is technically impossible without injecting BrdU into mice. Please see instead, a low number of Ki67-positive cells in the mouse kidney (Fig 6G). While BrdU marks exclusively cells that progressed from the G1 to S phase of the cell cycle, Ki67 marks cells in the G1, S, G2 and M phases. This means that a number of BrdU-positive cells is even less, which confirms a minimal proliferation in the kidney during homeostatic maintenance [8]. 

6. The observation that Wnt signaling controlled growth and differentiation of kidney organoids is very interesting. Why R-spondin-1 removal led to a switch from predominant cystic to fully solid organoids, while treatment with ICG-001 at IC50 did not result in morphological changes? The authors could try to provide and explanation and this should be discussed.

Thank you for this important question. We rule out a possibility that ICG-001 at IC50 did not sufficiently inhibit Wnt downstream comparing to the removal of the upstream agonist R-spondin-1, as we observed upregulation of some differentiation markers in both conditions (Fig 4I-L). We can only speculate that ICG-001, despite being a well-established Wnt inhibitor [25], also results by binding to Creb-binding protein (CBP) in off-target blockage of histone acetyltransferases, epigenetic regulators [26], which might interfere with the on-target phenotypes.

7. Sentences in lines 613-618 are not clear. Please revise.

We have now revised these sentences (see page 28 and 29, Revised Manuscript with Track Changes).

8. In the discussion, I suggest to remove paragraph headings in bold. The first heading “Tubular organoid cultures from adult mouse kidneys enrich for stem/progenitor cells with active Wnt and Notch signaling”, as well as the second one, are not correct for the reason explained in point 1.

We have now revised the first heading according to our explanations in point 1., i.e. Tubuloid cultures from adult mouse kidneys enrich for Wnt- and Notch-high cells with stem/progenitor-like functionality (see page 34, Revised Manuscript with Track Changes). Though, we have to disagree that the second heading is incorrect, as it relates to our previous study on sorted Wnt- and Notch-high kidney cancer stem cells from humans [4]. 

9. I suggest to focus the entire paper on the important role of Wnt and Notch that transiently activated can accelerate the regeneration but sustained activation can promote maladaptative behaviors and CKD progression. This is a very interesting point.

Thank you. We agree that these are important phenotypes, which already serve as a backbone of our manuscript. We have now slightly updated Discussion to emphasize the different responses even more (see page 37, Revised Manuscript with Track Changes). Though, we argue that modeling Wnt- and Notch-high human clear cell kidney cancer in mice was our initial goal, so it should remain the most important conceptual point.

Our rationale was based on our previous findings that Wnt and Notch signaling are elevated in CXCR4+MET+CD44+ cancer stem cells from primary human clear cell kidney tumors, and maintain self-renewal and tumorigenicity of these cells in relevant patient-derived systems [4]. To model Wnt- and Notch-dependent stem cell responses in kidney tumorigenesis in mice, we aimed first to examine the importance of Wnt and Notch for normal adult renal cells with stem/progenitor-like functionality in tubuloids. So far, organoid cultures are considered the most reliable and conclusive system to study stem cell properties in vitro [3,5,27]. According to the cancer stem cell hypothesis [28], normal stem/progenitor cells might be the cells of origin for Wnt- and Notch-high cancer stem cells in kidney tumors, once dysregulated. 

10. The role of Wnt and Notch signaling in normal human adult renal stem/progenitor cells that has been reported in several studies (Please see, PMID: 35922038, PMID: 37190024, PMID: 23861881, PMID: 19843711) should be also discussed.

Unfortunately, PMID: 35922038, PMID: 37190024 and PMID: 23861881 focus on the role of long non-coding RNAs and microRNAs in PROM1-positive adult human kidney stem/progenitor cells. However, these studies did not report the role of Wnt and Notch, and thus are not relevant for merit of our manuscript. Nevertheless, we have now incorporated PMID: 19843711 in the Introduction to point out that Wnt and Notch are active in PROM1-positive cells. We have also incorporated PMID: 29431914, one of the original studies cited in the review article PMID: 37190024, as it revealed a role of PROM1 in activating Wnt signaling in these cells (see page 3, Revised Manuscript with Track Changes). 

References

1. Nishinakamura R. Human kidney organoids: progress and remaining challenges. Nat Rev Nephrol. 2019;15: 613–624. doi:10.1038/s41581-019-0176-x

2. Schutgens F, Rookmaaker MB, Margaritis T, Rios A, Ammerlaan C, Jansen J, et al. Tubuloids derived from human adult kidney and urine for personalized disease modeling. Nat Biotechnol. 2019;37: 303–313. doi:10.1038/s41587-019-0048-8

3. Schutgens F, Clevers H. Human Organoids: Tools for Understanding Biology and Treating Diseases. Annu Rev Pathol Mech Dis. 2020;15: 211–234. doi:10.1146/annurev-pathmechdis-012419-032611

4. Fendler A, Bauer D, Busch J, Jung K, Wulf-Goldenberg A, Kunz S, et al. Inhibiting WNT and NOTCH in renal cancer stem cells and the implications for human patients. Nat Commun. 2020;11: 929. doi:10.1038/s41467-020-14700-7

5. Jensen KB, Little MH. Organoids are not organs: Sources of variation and misinformation in organoid biology. Stem cell reports. 2023;18: 1255–1270. doi:10.1016/j.stemcr.2023.05.009

6. Lassé M, El Saghir J, Berthier CC, Eddy S, Fischer M, Laufer SD, et al. An integrated organoid omics map extends modeling potential of kidney disease. Nat Commun. 2023;14: 4903. doi:10.1038/s41467-023-39740-7

7. Bussolati B, Bruno S, Grange C, Buttiglieri S, Deregibus MC, Cantino D, et al. Isolation of Renal Progenitor Cells from Adult Human Kidney. Am J Pathol. 2005;166: 545–555. doi:10.1016/S0002-9440(10)62276-6

8. Kang HM, Huang S, Reidy K, Han SH, Chinga F, Susztak K. Sox9-Positive Progenitor Cells Play a Key Role in Renal Tubule Epithelial Regeneration in Mice. Cell Rep. 2016;14: 861–871. doi:10.1016/j.celrep.2015.12.071

9. Sagrinati C, Netti GS, Mazzinghi B, Lazzeri E, Liotta F, Frosali F, et al. Isolation and Characterization of Multipotent Progenitor Cells from the Bowman’s Capsule of Adult Human Kidneys. J Am Soc Nephrol. 2006;17: 2443–2456. doi:10.1681/ASN.2006010089

10. Kumar S, Liu J, Pang P, Krautzberger AM, Reginensi A, Akiyama H, et al. Sox9 Activation Highlights a Cellular Pathway of Renal Repair in the Acutely Injured Mammalian Kidney. Cell Rep. 2015;12: 1325–1338. doi:10.1016/j.celrep.2015.07.034

11. Zhu L, Finkelstein D, Gao C, Shi L, Wang Y, López-Terrada D, et al. Multi-organ Mapping of Cancer Risk. Cell. 2016;166: 1132-1146.e7. doi:10.1016/j.cell.2016.07.045

12. Barker N, van Es JH, Kuipers J, Kujala P, van den Born M, Cozijnsen M, et al. Identification of stem cells in small intestine and colon by marker gene Lgr5. Nature. 2007;449: 1003–7. doi:10.1038/nature06196

13. Kobayashi A, Valerius MT, Mugford JW, Carroll TJ, Self M, Oliver G, et al. Six2 defines and regulates a multipotent self-renewing nephron progenitor population throughout mammalian kidney development. Cell Stem Cell. 2008;3: 169–81. doi:10.1016/j.stem.2008.05.020

14. Post Y, Clevers H. Defining Adult Stem Cell Function at Its Simplest: The Ability to Replace Lost Cells through Mitosis. Cell Stem Cell. 2019;25: 174–183. doi:10.1016/j.stem.2019.07.002

15. Kusaba T, Lalli M, Kramann R, Kobayashi A, Humphreys BD. Differentiated kidney epithelial cells repair injured proximal tubule. Proc Natl Acad Sci. 2014;111: 1527–1532. doi:10.1073/pnas.1310653110

16. Chen J, Chen J-K, Conway EM, Harris RC. Survivin Mediates Renal Proximal Tubule Recovery from AKI. J Am Soc Nephrol. 2013;24: 2023–2033. doi:10.1681/ASN.2013010076

17. Rinkevich Y, Montoro DT, Contreras-Trujillo H, Harari-Steinberg O, Newman AM, Tsai JM, et al. In Vivo Clonal Analysis Reveals Lineage-Restricted Progenitor Characteristics in Mammalian Kidney Development, Maintenance, and Regeneration. Cell Rep. 2014;7: 1270–1283. doi:10.1016/j.celrep.2014.04.018

18. Sallustio F, De Benedictis L, Castellano G, Zaza G, Loverre A, Costantino V, et al. TLR2 plays a role in the activation of human resident renal stem/progenitor cells. FASEB J. 2010;24: 514–25. doi:10.1096/fj.09-136481

19. Zhou D, Li Y, Lin L, Zhou L, Igarashi P, Liu Y. Tubule-specific ablation of endogenous β-catenin aggravates acute kidney injury in mice. Kidney Int. 2012;82: 537–547. doi:10.1038/ki.2012.173

20. Fujii M, Matano M, Toshimitsu K, Takano A, Mikami Y, Nishikori S, et al. Human Intestinal Organoids Maintain Self-Renewal Capacity and Cellular Diversity in Niche-Inspired Culture Condition. Cell Stem Cell. 2018;23: 787-793.e6. doi:10.1016/j.stem.2018.11.016

21. Sato T, van Es JH, Snippert HJ, Stange DE, Vries RG, van den Born M, et al. Paneth cells constitute the niche for Lgr5 stem cells in intestinal crypts. Nature. 2011;469: 415–418. doi:10.1038/nature09637

22. Snippert HJ, van der Flier LG, Sato T, van Es JH, van den Born M, Kroon-Veenboer C, et al. Intestinal Crypt Homeostasis Results from Neutral Competition between Symmetrically Dividing Lgr5 Stem Cells. Cell. 2010;143: 134–144. doi:10.1016/j.cell.2010.09.016

23. Forbes JM. Mitochondria-Power Players in Kidney Function? Trends Endocrinol Metab. 2016;27: 441–442. doi:10.1016/j.tem.2016.05.002

24. Pan X. The Roles of Fatty Acids and Apolipoproteins in the Kidneys. Metabolites. 2022;12: 462. doi:10.3390/metabo12050462

25. Kahn M. Can we safely target the WNT pathway? Nat Rev Drug Discov. 2014;13: 513–32. doi:10.1038/nrd4233

26. Cheng Y, He C, Wang M, Ma X, Mo F, Yang S, et al. Targeting epigenetic regulators for cancer therapy: mechanisms and advances in clinical trials. Signal Transduct Target Ther. 2019;4: 62. doi:10.1038/s41392-019-0095-0

27. Lancaster MA, Knoblich JA. Organogenesis in a dish: Modeling development and disease using organoid technologies. Science. 2014;345: 1247125. doi:10.1126/science.1247125

28. Nassar D, Blanpain C. Cancer Stem Cells: Basic Concepts and Therapeutic Implications. Annu Rev Pathol Mech Dis. 2016;11: 47–76. doi:10.1146/annurev-pathol-012615-044438

---

## [Decision Letter · Decision Letter 1]

2 Oct 2023

PONE-D-23-05670R1Mice with renal-specific alterations of stem cell-associated signaling develop symptoms of chronic kidney disease but surprisingly no tumorsPLOS ONE

Dear Dr. Myszczyszyn,

Thank you for submitting your manuscript to PLOS ONE. After careful consideration, we feel that it has merit but does not fully meet PLOS ONE’s publication criteria as it currently stands. Therefore, we invite you to submit a revised version of the manuscript that addresses the points raised during the review process.

I agree with the reviewer 2 about the headings in the discussion, please removed - and address (again) their point about figure 2D.  With these changes the ms. is likely to acceptable without re-review.  

We look forward to receiving your revised manuscript.

Kind regards,

Michael Klymkowsky, Ph.D.

Academic Editor

PLOS ONE

Journal Requirements:

Reviewers' comments:

Reviewer's Responses to Questions

**Comments to the Author**

1. If the authors have adequately addressed your comments raised in a previous round of review and you feel that this manuscript is now acceptable for publication, you may indicate that here to bypass the “Comments to the Author” section, enter your conflict of interest statement in the “Confidential to Editor” section, and submit your "Accept" recommendation.

Reviewer #1: All comments have been addressed

Reviewer #2: (No Response)

2. Is the manuscript technically sound, and do the data support the conclusions?

Reviewer #1: Yes

Reviewer #2: Partly

3. Has the statistical analysis been performed appropriately and rigorously? 

Reviewer #1: Yes

Reviewer #2: Yes

4. Have the authors made all data underlying the findings in their manuscript fully available?

Reviewer #1: Yes

Reviewer #2: Yes

5. Is the manuscript presented in an intelligible fashion and written in standard English?

Reviewer #1: Yes

Reviewer #2: Yes

6. Review Comments to the Author

Reviewer #1: (No Response)

Reviewer #2: Thank you for your revisions.

The paper has now improved even if I think there is stil the point 5 to address.

Even after the author explanation, The Figure 2D remains misleading since even if the aim of Fig 2D was the functional confirmation of a high number of proliferating cells in tubuloids, the comparison between BrdU cells derived from organoids and EPCAM+ and adult kidney epithelial cells (control kidney) should be shown. I suggest eliminating Fig 2D and that to support the assertion that Kidney tubuloids displayed enhanced proliferation showing a new graphic from their proteomic data showing the expression of proliferation markers Ki67, Pcna and Cyclin D1 in the three conditions.

7. PLOS authors have the option to publish the peer review history of their article (what does this mean?). If published, this will include your full peer review and any attached files.

Reviewer #1: No

Reviewer #2: No

---

## [Author Response · Author response to Decision Letter 1]

20 Dec 2023

Berlin, November 16th, 2023

Rebuttal letter

Dear dr. Klymkowsky,

We would like to thank you and both reviewers for the efforts in assessing our revised manuscript. We are happy to hear that the manuscript will likely be accepted upon a few minor changes. 

Editor:

I agree with the reviewer 2 about the headings in the discussion, please removed - and address (again) their point about figure 2D. With these changes the ms. is likely to acceptable without re-review. 

Reviewer 2:

The paper has now improved even if I think there is still the point 5 to address. Even after the author explanation, The Figure 2D remains misleading since even if the aim of Fig 2D was the functional confirmation of a high number of proliferating cells in tubuloids, the comparison between BrdU cells derived from organoids and EPCAM+ and adult kidney epithelial cells (control kidney) should be shown. I suggest eliminating Fig 2D and that to support the assertion that kidney tubuloids displayed enhanced proliferation showing a new graphic from their proteomic data showing the expression of proliferation markers Ki67, Pcna and Cyclin D1 in the three conditions.

We have now fully addressed these points. 

We look forward to hearing from you and would be glad to respond to any further questions or comments.

Yours sincerely,

Dr. Adam Myszczyszyn, first author

Prof. Dr. Walter Birchmeier, senior author

---

## [Decision Letter · Decision Letter 2]

14 Jan 2024

Mice with renal-specific alterations of stem cell-associated signaling develop symptoms of chronic kidney disease but surprisingly no tumors

PONE-D-23-05670R2

Dear Dr. Myszczyszyn,

We’re pleased to inform you that your manuscript has been judged scientifically suitable for publication and will be formally accepted for publication once it meets all outstanding technical requirements.

Kind regards,

Michael Klymkowsky, Ph.D.

Academic Editor

PLOS ONE

Additional Editor Comments (optional):

Reviewers' comments:

Reviewer's Responses to Questions

**Comments to the Author**

1. If the authors have adequately addressed your comments raised in a previous round of review and you feel that this manuscript is now acceptable for publication, you may indicate that here to bypass the “Comments to the Author” section, enter your conflict of interest statement in the “Confidential to Editor” section, and submit your "Accept" recommendation.

Reviewer #2: All comments have been addressed

2. Is the manuscript technically sound, and do the data support the conclusions?

Reviewer #2: Partly

3. Has the statistical analysis been performed appropriately and rigorously? 

Reviewer #2: Yes

4. Have the authors made all data underlying the findings in their manuscript fully available?

Reviewer #2: No

5. Is the manuscript presented in an intelligible fashion and written in standard English?

Reviewer #2: Yes

6. Review Comments to the Author

Reviewer #2: (No Response)

7. PLOS authors have the option to publish the peer review history of their article (what does this mean?). If published, this will include your full peer review and any attached files.

Reviewer #2: No

---

## [Editor Report · Acceptance letter]

11 Mar 2024

PONE-D-23-05670R2 

PLOS ONE

Dear Dr. Myszczyszyn, 

I'm pleased to inform you that your manuscript has been deemed suitable for publication in PLOS ONE. Congratulations! Your manuscript is now being handed over to our production team.

Kind regards, 

on behalf of

Dr. Michael Klymkowsky 

Academic Editor

PLOS ONE